# Modelling ozone-induced changes in wheat amino acids and protein quality using a process-based crop model

Jo Cook[1], Durgesh Singh Yadav[2], Felicity Hayes[3], Nathan Booth[1], Sam Bland[4], Pritha Pande[4], Samarthia Thankappan[1], Lisa Emberson[1]

[1]Environment & Geography Dept., University of York, YO10 5DD, UK
[2]Department of Botany, Government Raza P.G. College, Rampur, India
[3]UK Centre for Ecology & Hydrology, Environment Centre Wales, Bangor, Wales, UK
[4]Stockholm Environment Institute at York, Environment & Geography Dept., University of York, YO10 5DD, UK

*Correspondence to*: Jo Cook (jo.cook@york.ac.uk)

**Abstract.** Ozone ($O_3$) pollution reduces wheat yields as well as the protein and micronutrient yield of the crop. $O_3$ concentrations are particularly high in India, and are set to increase, threatening wheat yields and quality in a country already facing challenges to food security. This study aims to improve the existing $DO_3SE$-CropN model to simulate the effects of $O_3$ on Indian wheat quality by incorporating antioxidant processes to simulate protein, and the concentrations of nutritionally relevant amino acids. As a result, the improved model can now capture the decrease in protein concentration that occurs in Indian wheat exposed to elevated $O_3$. The structure of the modelling framework is transferrable to other abiotic stressors and easily integrable into other crop models, provided they simulate leaf and stem N, demonstrating the flexibility and usefulness of the framework developed in this study. Further, the modelling results can be used to simulate the FAO recommended metric for measuring protein quality, the DIAAS, setting up a foundation for nutrition-based risk assessments of $O_3$ effects on crops. The resulting model was able to capture grain protein, lysine and methionine concentrations reasonably well. As a proportion of dry matter, the simulated percentages ranged from 0.26% to 0.38% for lysine, and from 0.13% to 0.22% for methionine, while the observed values were 0.16% to 0.38% and 0.14% to 0.22%, respectively. For grain and leaf protein simulations, the interdependence between parameters reduced the accuracy of their respective relative protein loss under $O_3$ exposure. Additionally, the decrease in lysine and methionine concentrations under $O_3$ exposure was underestimated by ~10 percentage points for methionine for both cultivars, and by 37 and 19 percentage points for lysine for HUW234 and HD3118 respectively. This underestimation occurs despite simulations of relative yield loss being fairly accurate (average deviation of 2.5 percentage points excluding outliers). To provide further mechanistic understanding of $O_3$ effects on wheat grain quality, future experiments should measure nitrogen (N) and protein concentrations in leaves and stems, along with the proportion of N associated with antioxidants, which will aid in informing future model development. Additionally, exploring how grain protein relates to amino acid concentrations under $O_3$ will enhance the model's accuracy in predicting protein quality and provide more reliable estimates of the influence of $O_3$ on wheat quality. This study builds on the work of Cook et al. (2024) and supports the

second phase of the tropospheric $O_3$ assessment report (TOAR) by investigating the impacts of tropospheric $O_3$ on Indian wheat and the potential of this to exacerbate existing malnutrition in India.

## 1. Introduction

A growing body of literature from Europe, China and India has shown that exposure to $O_3$ reduces wheat protein and micronutrient yields (Broberg et al., 2015; Feng et al., 2008; Mishra et al., 2013; Yadav et al., 2020). This is important as cereals often make up the most available protein source per capita and wheat is the dominant dietary cereal globally (Shiferaw et al., 2013). Therefore, any reduction in yield, protein and micronutrient contents caused by $O_3$ could threaten both food and nutrition security, especially in countries such as India where $O_3$ concentrations are high and food security is low (FAO et al.,

2020; Herforth et al., 2020; Mills et al., 2018b). The first phase of the tropospheric ozone ($O_3$) assessment report (TOAR) (https://igacproject.org/activities/TOAR/TOAR-I) compiled information on surface $O_3$ metrics to produce the world's largest database for identification of global distribution and trends of $O_3$ (Schultz et al., 2017). From the first phase of TOAR, it was observed that tropospheric $O_3$ increased globally in the 20[th] century, with atmospheric chemistry and climate modelling studies finding that $O_3$ production is greatest in mid to high latitudes due to greater emissions of $O_3$ pre-cursors (Archibald et al., 2020;

Cooper et al., 2014). Additionally, using the database Mills et al. (2018b) found that in East Asia $O_3$ concentration metrics for wheat growing locations were much greater than in Europe. Several authors for the first phase of TOAR commented on the underrepresentation of some key wheat producing areas (particularly India but also for China and Russia) in the database, which limited some of the analysis (Cooper et al., 2014; Mills et al., 2018b; Schultz et al., 2017). This paper is part of the second phase of TOAR (https://igacproject.org/activities/TOAR/TOAR-II), which expands on the first phase to investigate $O_3$

impacts on human health and vegetation. This study contributes to the second phase of TOAR by examining the impacts of tropospheric $O_3$ on wheat yield and quality in India, enhancing our understanding of the broader implications for food and nutrition security. Understanding the interplay of different factors affecting $O_3$ induced reductions in wheat yield and quality will be important for current, and future, food and nutritional security risk assessments.

### 1.1 Malnutrition and the importance of wheat in India

Malnutrition is prevalent in India with ~40% of the population unable to afford a nutritionally adequate diet, and ~80% unable to afford a healthy diet (FAO et al., 2023). In India, ~35% of children under the age of 5 are affected by stunting and ~20% affected by wasting, with the prevalence of wasting in India being one of the highest in the world (Global Nutrition Report | Country Nutrition Profiles - Global Nutrition Report). Stunting and wasting occur when an individual does not have sufficient calories or micronutrients in their diet to grow and develop (Gonmei and Toteja, 2018). Wasting and muscle function loss can

result from a poor quantity, or quality, of dietary protein (Medek et al., 2017). For most Indian states, at least 30% of the population are at risk of protein deficiency, which is of concern for people who are pregnant or in poorer socioeconomic

circumstances, who require higher qualities of protein for growth or fighting infections (Minocha et al., 2017; Swaminathan et al., 2012).

In India, cereals are the most available protein source per capita, and are a key dietary protein source (Minocha et al., 2017).
Wheat makes up the dominant dietary cereal in the north of India where the majority of the crop is grown (Khatkar et al., 2015). Globally, India has the greatest area under wheat cultivation, 31.6 million hectares, and produced 109.5 million tonnes of wheat in 2021, second only to the amount of wheat produced by China (Ministry of Agriculture & Farmers Welfare, 2022). As a result, the country is self-sufficient/reliant when it comes to wheat (Tripathi and Mishra, 2017). Consumption of wheat varies by state with the dominant wheat producing states (Punjab, Rajasthan, Haryana and Madhya Pradesh) consuming the
most. Resulting from population growth and increases to income, the total demand for wheat is increasing (Tripathi and Mishra, 2017). However, numerous experimental and modelling studies have shown that $O_3$ is substantially reducing wheat yields across India (Mills et al., 2018a; Mishra et al., 2013; Sharma et al., 2019; Sinha et al., 2015; Yadav et al., 2021).

## 1.2 $O_3$ pollution in India

Ground level $O_3$ is a secondary pollutant, formed when precursor gases (predominantly volatile organic compounds and
nitrogen oxides) react in the presence of ultraviolet light (Fowler et al., 2008). The first phase of TOAR, identified that South Asia, and in particular India, experience some of the highest $O_3$ burdens of any region or country worldwide, though this analysis was limited by the availability of $O_3$ concentration data for India (Emberson, 2020; Mills et al., 2018b). These high $O_3$ burdens occur due to increasing pre-cursor emissions and insufficient pollution control measures (Archibald et al., 2020; Elshorbany et al., 2024; Singh et al., 2023; Wang et al., 2023). Atmospheric chemistry and climate models have found that
geographically, the highest $O_3$ concentrations in India occur in the northern part of the country and the Indo-Gangetic planes (IGP), where the majority of wheat is grown (Lu et al., 2018; Ministry of Agriculture & Farmers Welfare, 2022; Rathore et al., 2023). In the future, the changing climate will affect $O_3$ concentrations, with model projections agreeing that climatic conditions across the north of India will favour greater $O_3$ production (Kumar et al., 2018; Li et al., 2022a; Stevenson et al., 2013). Using a Nested Regional Climate Model with Chemistry, Kumar et al. (2018) projected that $O_3$ concentrations across
India will rise under RCP 8.5, while remaining comparable to current levels under RCP 6.0. For the dry, wheat growing season, the authors projected that $O_3$ concentrations across the IGP will increase under both RCP 6.0 and RCP 8.5, with a much larger increase under RCP 8.5 (Kumar et al., 2018). This is a critical finding given the majority of wheat is grown in the north of India, across the IGP (Ministry of Agriculture & Farmers Welfare, 2022).

## 1.3 Effects of $O_3$ pollution on wheat yields

$O_3$ diffuses into wheat leaves via the stomates and impacts photosynthesis and senescence when antioxidant defences are compromised (Emberson et al., 2018; Rai and Agrawal, 2012; Tiwari and Agrawal, 2018). Accelerated senescence shortens the grain filling period, and the decline in photosynthesis decreases biomass production, ultimately leading to lower crop yields (Emberson et al., 2018; Tiwari and Agrawal, 2018).

Several experimental studies using wheat cultivars commonly grown in India have shown decreases in yield due to elevated $O_3$ exposure (Naaz et al., 2022; Pandey et al., 2018; Tomer et al., 2015; Yadav et al., 2021). National estimates of relative yield (RY) loss due to $O_3$ across India vary between 3.8-41% between studies (Avnery et al., 2011; Van Dingenen et al., 2009; Droutsas, 2020; Ghude et al., 2014; Lal et al., 2017; Sharma et al., 2019; Sinha et al., 2015). The effects of $O_3$ on wheat yield also vary spatially (Droutsas, 2020; Ghude et al., 2014; Lal et al., 2017; Mills et al., 2018a; Sharma et al., 2019). Mills et al. (2018) found the greatest yield losses across the north of the country as the meteorological conditions are more favourable to $O_3$ uptake. Lal et al. (2017) also found the greatest wheat yield losses due to $O_3$ in the north and west of India, where the majority of wheat is grown. Further, Naaz et al. (2022) exposed Indian wheat cultivars to different conditions representing future $O_3$ and climate scenarios, finding that areas suitable for wheat cultivation will be reduced in the future.

**1.4 Effects of $O_3$ pollution on wheat quality**

Studies have shown that the starch, protein and micronutrient yield of wheat decreases under elevated $O_3$ exposure (Broberg et al., 2015; Piikki et al., 2008; Tomer et al., 2015). Pre-anthesis, the accumulation of nitrogen (N) in upper plant parts is unaffected by elevated $O_3$ concentrations (Brewster et al., 2024). However, after anthesis, the $O_3$ induced acceleration of plant senescence limits the remobilisation of N from the leaves and stem to the grain (Brewster et al., 2024; Broberg et al., 2017; Chang-Espino et al., 2021). Brewster, Fenner and Hayes (2024) also suggest that an additional process affects N remobilisation to the grain, as they found an increase in residual N in the flag leaf, despite not detecting a difference in senescence onset. It is possible that the residual N is in the form of antioxidants (for example glutathione) which the plant creates for defence against $O_3$ induced reactive oxygen species (ROS) (Brewster et al., 2024; Sarkar and Agrawal, 2010; Yadav et al., 2019). Overall, the reduction in N remobilisation, leads to reduced N deposition to the grain and a reduced grain N, and protein, yield (Broberg et al., 2015; Cook et al., 2024; Yadav et al., 2020).

In wheat, since the grain yield is decreased to a greater extent than proteins and micronutrients under increased $O_3$, the concentration of protein and micronutrients in the grains generally increases (Feng and Kobayashi, 2009; Piikki et al., 2008). However, some wheat varieties, particularly Indian wheat, have shown a different pattern, where the protein yield and concentration of the grains decreases under $O_3$ exposure (Baqasi et al., 2018; Mishra et al., 2013; Yadav et al., 2020).

Indispensable amino acids (AA's) are most important for nutrition as they cannot be synthesised by the body and must be obtained through diet (Brestenský et al., 2019). Additionally, the quantity of N containing compounds consumed is important for synthesis of dispensable AA (Brestenský et al., 2019). Nevertheless, while dispensable AA can be produced by the body, their consumption is still important for supporting metabolic functions (Brestenský et al., 2019). The production of different proteins in the body requires AA in differing proportions (Shewry and Hey, 2015). The AA that is available in the lowest proportion, the most limiting AA, determines protein production (Elango et al., 2008; Shewry and Hey, 2015). Un-utilised AA cannot be stored so if they are not used for protein production they are oxidised (Brestenský et al., 2019; Elango et al., 2008). Yadav et al. (2020) looked at the AA profiles of a modern (HD3118), and old (HUW234), wheat cultivar exposed to $O_3$,

finding indispensable and dispensable AA decreased under $O_3$ exposure. The effect of $O_3$ on protein quality of wheat is of particular concern given the existing state of malnutrition in India.

## 1.5 Crop modelling for $O_3$ and nutrition

Several crop models have been used to investigate the impacts of $O_3$ pollution on crop yields in a wide range of countries and globally (Droutsas, 2020; Guarin et al., 2019, 2024; Nguyen et al., 2024; Schauberger et al., 2019; Tai et al., 2021; Tao et al., 2017; Tian et al., 2015; Xu et al., 2023; Zhou et al., 2018). Ebi et al. (2021) highlight the usefulness of models for such risk assessments, while stressing that most do not consider aspects relevant for human nutrition in their simulations. Currently, only one model has been developed which captures the effect of $O_3$ on crop nutrition: $DO_3SE$-CropN (Cook et al., 2024). $DO_3SE$-CropN is built on the existing $DO_3SE$-Crop model, which takes inputs of hourly meteorology and $O_3$ concentrations to simulate crop phenology, $O_3$-impacted net photosynthesis, dry matter partitioning, grain filling and $O_3$ impacted crop senescence (Pande et al., 2024a). The $DO_3SE$-CropN model then simulates crop N, and models explicitly the effect of $O_3$ on reducing the amount of N from the leaves and stems that is available for the grain. From the grain N content (gN m$^{-2}$), grain protein content (gProtein m$^{-2}$) is easily obtained using conversion factors (Mariotti et al., 2008).

The $DO_3SE$-CropN model was originally developed to capture the increase in N concentration (100gN gDM$^{-1}$) and decrease in N yield (gN m$^{-2}$) that occurs under $O_3$ exposure in European wheat (Cook et al., 2024). However, Indian wheat experiences a decrease in grain protein concentration as well as a decrease in grain protein yield under elevated $O_3$ concentrations (Mishra et al., 2013; Yadav et al., 2020). In India, the ambient $O_3$ concentrations are high, leading to ROS production and subsequent yield losses (Sharma et al., 2019; Sinha et al., 2015; Tiwari and Agrawal, 2018). The production of antioxidants by the plant to defend against ROS reduces the proportion of proteins that would otherwise be remobilised to the grain, reducing grain protein (Yadav et al., 2019, 2020). Therefore, to capture the decrease in the protein concentration of Indian wheat under $O_3$ exposure, the inclusion of antioxidant processes is essential. Further, the inclusion of such processes will improve the wider applicability of the model for simulating $O_3$ effects on wheat quality for regions with high $O_3$ concentrations.

Further, to expand the nutritional relevance of the model, it would be useful to simulate the effect of $O_3$ on protein quality. This can be done through simulating AA concentrations, which can subsequently be used to calculate the recommended metric for measuring protein quality by the FAO: the dietary indispensable AA score (DIAAS). The inclusion of protein quality would allow for risk assessments of $O_3$ effects on wheat nutrition in addition to yield.

## 1.6 Aims

In the present study, the $DO_3SE$-CropN model was further developed and applied with two years of meteorological data. The model was calibrated using phenology, photosynthesis and yield data collected for two cultivars (HUW234 and HD3118) grown under both ambient and elevated $O_3$ treatments. All data were available from Yadav, Agrawal and Agrawal (2021). Grain quality data were obtained from an experiment on the same cultivars a year prior, however hourly meteorological and $O_3$ data were not available for this year (Yadav et al., 2020). In the absence of further data, this study assumes that the grain

protein concentration and grain protein quality will respond similarly to $O_3$ between years. The aims of the present study were to use the available data for the following:

1) Develop a framework to simulate the antioxidant response of wheat under $O_3$ exposure for incorporation into the existing DO$_3$SE-CropN model
2) Develop a method for simulating the impact of $O_3$ exposure on the protein quality of wheat, focussing on AA essential for human nutrition, for incorporation into the existing DO$_3$SE-CropN model.

## 2 Model development

### 2.1 Integrating antioxidant processes into DO$_3$SE-CropN

The DO$_3$SE-Crop model is a coupled stomatal conductance-photosynthesis model, which simulates stomatal $O_3$ uptake and its impact on photosynthesis which the plant can recover from overnight, as well as $O_3$ induced accelerated crop senescence (Pande et al., 2024a). Daily photosynthate is partitioned between the leaves, stem, roots and grains according to the plant's growth stage (Osborne et al., 2015; Pande et al., 2024a). Development of DO$_3$SE-Crop has allowed the $O_3$ impact on wheat

production in China and Europe to be estimated (Nguyen et al., 2024; Pande et al., 2024a). The N module for DO$_3$SE-Crop, developed by Cook et al. (2024), takes inputs of daily stem and leaf dry matter (DM), as well as the onset of crop senescence, to simulate the N accumulated by the leaf and stem. The remobilisation of N from the leaf and stem to the grain after anthesis is simulated using a sigmoid function. To account for the reduction in N remobilisation under $O_3$ exposure, a relationship linking accumulated $O_3$ flux to the minimum N levels in the leaf and stem is incorporated (Brewster et al., 2024; Cook et al.,

2024). The model allowed the decrease in grain N yield (gN m$^{-2}$) and increase in grain N concentration (100gN gDM$^{-1}$) of European wheat under $O_3$ exposure to be simulated (Cook et al., 2024). A full write up of the equations and processes of the DO$_3$SE-Crop model is given in Pande et al. (2024). Additionally, a full description of the equations and processes of version 1.0 of the N module developed for DO$_3$SE-Crop is given in (Cook et al., 2024). In this study version 4.39.16 of the DO$_3$SE-Crop model was used (Bland, 2024), along with version 2.0 of the N module (Cook, 2024).

The first iteration of the N module for DO$_3$SE-Crop (Cook et al., 2024), did not consider the utilisation of leaf/stem N in creating defence proteins, yet for Indian wheat this may be an important process to explain the decrease in grain protein concentration as well as yield (Yadav et al., 2019, 2020). Here we propose a method by which the leaf and stem N involved in antioxidant production may be quantified (Fig. 1). For the purposes of this study, we do not consider individual antioxidants (e.g. superoxide dismutase (Tiwari and Agrawal, 2018)). Instead, we model a general pool of N that we hypothesise to be

associated with antioxidants. This antioxidant pool of N is subsequently unavailable to the grain and is suggested to partially explain the decrease in grain protein of Indian wheat under $O_3$ exposure.

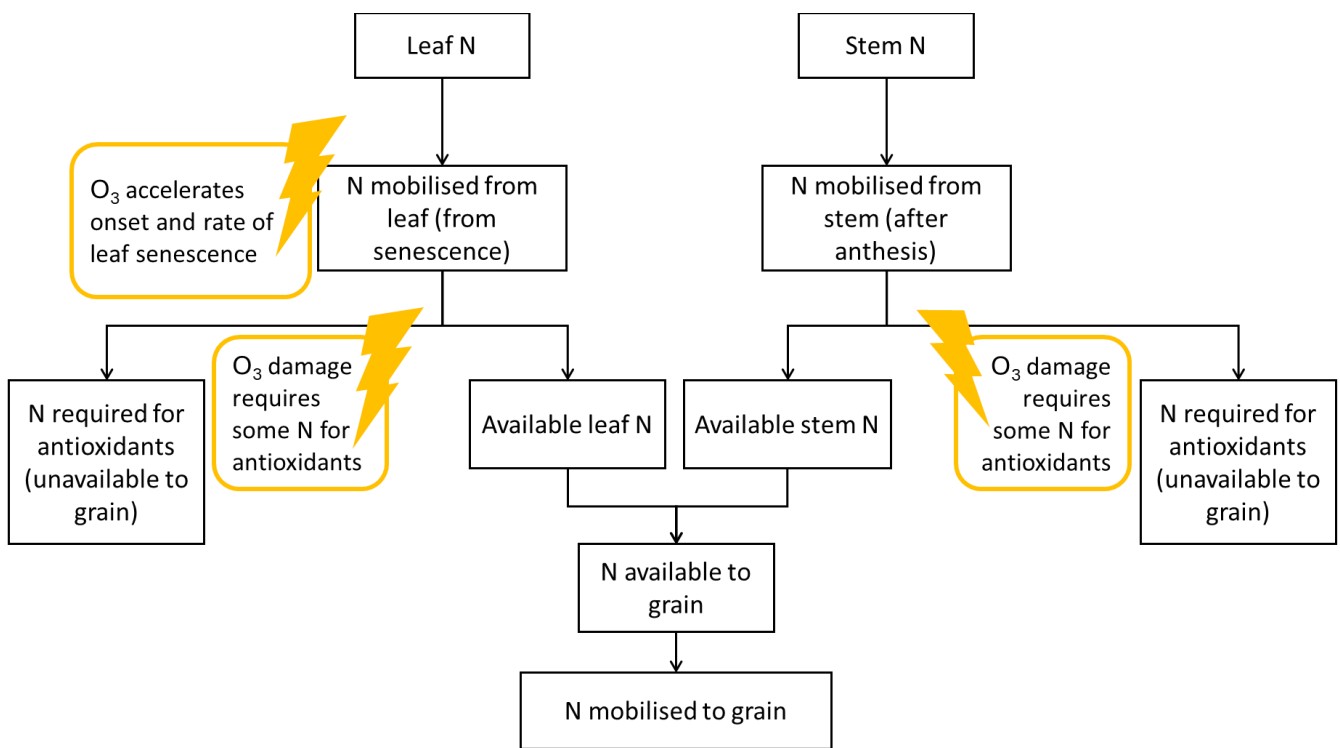

**Fig. 1: Diagram of the proposed method for integrating antioxidant response under O₃ exposure into the existing N module for DO₃SE-Crop. The lightning strikes represent where O₃ can interrupt plant N processes. The lightning strikes represent the points where O₃ interacts with the antioxidant processes in the model.**

Fig. 1 shows how antioxidant processes can be integrated within the existing DO₃SE-CropN framework. When the leaf senesces, N is released from the leaf. The N module is linked to the existing DO₃SE-Crop model so that increasing stomatal O₃ flux accelerates senescence, which accelerates N release from the leaf. N is released until the minimum leaf N concentration is reached. Previously, the minimum leaf N concentration increased with O₃ concentration to represent the increase in residual N (Cook et al., 2024). Now it is hypothesised that this increase in residual N is due to the leaf and stem using N for antioxidants which remain in the leaf. After a threshold of accumulated O₃ flux has been exceeded, we allocate a proportion of the released N to an antioxidant pool, which means it is unavailable to the grain. Since the stem is also involved in antioxidant response and defence against ROS (Bazargani et al., 2011; Gao et al., 2018; Li et al., 2022b), the same mechanism is used for the stem. We determine the proportion of N that will be allocated to the antioxidant pool using an equation that follows a similar structure as the drought stress factor of Liu et al. (2018), as both O₃ and drought stress are ROS mediated (Khanna-Chopra, 2012). Liu et al. (2018) use their drought stress factor to empirically modify the N:protein conversion factor under drought stress. Here, we introduce this method to the DO₃SE-CropN model via equation (1) as a more mechanistic approach. Instead of modifying the N:protein conversion factor under an abiotic stress, we use the structure of Liu et al.'s (2018) equation to determine N allocation to the antioxidant pool, thereby reducing the N available to the grain, and subsequently affecting grain protein.

The proportion of N allocated to the antioxidant pool, $f_{O_3,Antioxidants}$, takes the following form:

$$f_{O_3,Antioxidants} = \begin{cases} 0, & fst_{acc} < cL_{O_3} \\ \dfrac{fst_{acc} - cL_{O_3}}{a_{part} \times fst_{end} - cL_{O_3}}, & fst_{acc} \geq cL_{O_3} \end{cases} \tag{1}$$

where $fst_{acc}$ is the current stomatal accumulation of $O_3$ flux in the DO₃SE model, $fst_{end}$ is the stomatal accumulation of $O_3$ flux when N is only allocated to antioxidant pool and is not available to the grain, $cL_{O_3}$ is the critical level above which $O_3$ flux starts affecting the onset of senescence in the DO₃SE-Crop model, and $a_{part}$ is a constant modifier that can be calibrated to customise the $O_3$ effect on antioxidants for each plant part (leaf and stem). $a_{part}$ must be equal to or greater than $\frac{cL_{O_3}}{fst_{end}}$ for

the antioxidant factor equation to show a decrease in released N with accumulated $O_3$ flux. Further, $fst_{end}$ must be greater than $cL_{O_3}$. Of the N released that day, the proportion available to the grain is $1 - f_{O_3,Antioxidants}$. The $cL_{O_3}$ term was chosen as the $O_3$ stress factor as if $O_3$ has exceeded a critical threshold and is affecting senescence onset, we can hypothesise that the allocation of N to antioxidants to protect against $O_3$ stress will be increased. $fst_{end}$ was incorporated into the equation to allow the end point of the slope to be customised. For varying values of $a_{part}$, the $O_3$ stress factor is used to calculate the proportion

of N available to the grain as a function of accumulated stomatal $O_3$ flux according to Fig. 2.

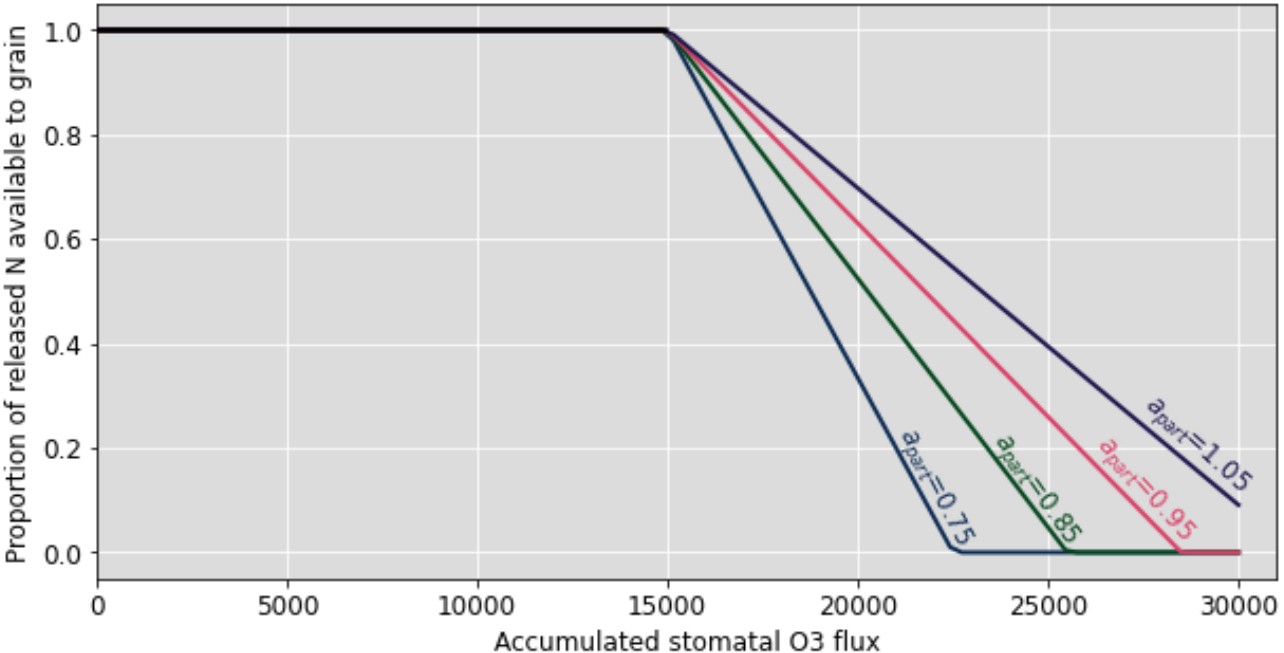

Fig. 2: Proportion of N released from stem or leaf senescence that is available to the grain for varying values of $a_{part}$. The plot uses $fst_{end} = 30000$, and $cL_{O_3} = 15000$.

**2.2 Identification of nutritionally relevant AA for O$_3$ exposed wheat**

The quality of a protein depends on the proportions of indispensable and dispensable AA's in the food. While Yadav et al. (2020) found that dispensable AA's were reduced to a greater extent than indispensable AA's under O$_3$ exposure, the most limiting for protein production were the indispensable AA's lysine and methionine. Additionally, the concentrations of lysine and methionine were reduced under O$_3$ exposure for both the HD3118 and HUW234 cultivars (Yadav et al., 2020). Therefore, to simulate the protein quality under O$_3$ exposure, lysine and methionine were focussed on.

**2.3 Protein and AA calculations**

The DO$_3$SE-CropN model outputs a grain N yield (gN m$^{-2}$) and concentration (100gN gDM$^{-1}$). From the grain N content, the protein content can be calculated by considering a standard N:protein conversion factor. The Jones' factors are commonly used to convert from N to protein, however, these factors vary between foods and within the same food group (Jones, 1941; Mariotti et al., 2008). On average for whole wheat, the conversion factor is 5.49, which is used in this study to convert grain N to protein

(Mariotti et al., 2008). The regressions used to calculate lysine and methionine concentrations of the wheat grain from grain protein percentage are taken from Table 5 of Liu et al. (2019).

**2.4 The DIAAS score**

The metric recommended by the FAO for evaluating food protein quality is the dietary indispensable AA score (DIAAS), which corrects for the AA digestibility at the end of the small intestine (FAO, 2013). It therefore reflects the fact that the

nutritional quality of protein should account for the AA required for metabolism (FAO, 2013). The metric also varies for different age groups which have different protein quality requirements (FAO, 2013). Currently, no crop model has incorporated a nutrition measure such as the DIAAS into their models. Additionally, no model has considered the impact of O$_3$ pollution on protein quality which is critical for risk assessments of O$_3$ stress on food and nutrition security.

There are two steps in calculating the DIAAS score. First, the DIAAS reference ratio is calculated for each AA as follows:

$$\genfrac{}{}{0pt}{}{DIAAS}{reference\ ratio} = \frac{True\ ileal\ IAA\ Digestibility \times mg\ of\ AA\ in\ 1g\ of\ the\ dietary\ protein}{mg\ of\ digestible\ indispensible\ AA\ in\ 1g\ of\ the\ dietary\ protein} \qquad (2)$$

where IAA stands for indispensable AA.

Once the AA concentrations have been obtained from grain protein simulations, as detailed in Section 2.3, then Eq. (2) is re-written using the parameters used in the crop modelling as:

$$\genfrac{}{}{0pt}{}{DIAAS}{reference\ ratio} = \frac{True\ ileal\ IAA\ Digestibility \times \dfrac{1000 \times Grain\ AA\ (\%\ in\ DM)}{Grain\ protein\ (\%\ in\ DM)}}{mg\ of\ digestible\ indispensible\ AA\ in\ 1g\ of\ the\ dietary\ protein} \qquad (3)$$

In the second step, the lowest DIAAS reference ratio is selected and used to calculate the DIAAS score as in Eq. (4). The

lowest reference ratio is selected as this corresponds to the AA which is most limiting in the food, and is available in the

smallest proportion relative to a person's requirements (Elango et al., 2008). The AA with the lowest availability determines protein production, and quality, and the other AA which are in excess of the most limiting one will be oxidised (Elango et al., 2008).

$$\frac{DIAAS}{score} = 100 \times lowest(reference\ ratio) \tag{4}$$

The true ileal IAA digestibility coefficients for wheat flour required for Eq.(3) can be obtained from Shaheen et al. (2016) for
the different AA. Additionally, the mg of digestible indispensable AA in 1g of the dietary protein are tabulated for the different AA and age groups in FAO (2013). There are different requirements for different age groups as adults only require AA for maintenance, whereas children require them for growth and maintenance (Shewry and Hey, 2015).

## 2.5 Calculations of RY loss, and the decrease in protein and AA concentrations under O₃

For performing risk assessments of $O_3$ damage to crops, RY and RY loss (1-RY) are the commonly used response parameters, which quantify the magnitude of the crop yield loss under $O_3$ by comparing it to the corresponding pre-industrial value (~10 ppb) (see Eq. 5) (CLRTAP, 2017). Such risk assessments allow for the magnitude of the effects of $O_3$ on crop yields to be estimated (Emberson, 2020).

$$Relative\ yield\ (RY) = \frac{Yield\ under\ O_3\ treatment}{Yield\ under\ preindustrial\ O_3} \tag{5}$$

For the model simulations, the yield under preindustrial $O_3$ was extracted by performing a model run with a constant $O_3$
concentration of 10 ppb, while the yields under the $O_3$ treatment were obtained by running the model with the hourly experimental $O_3$ concentration data for the ambient and elevated (ambient + 20 ppb) treatments. To extract the yield under preindustrial $O_3$ concentrations for the experimental data, the yields for the ambient and elevated treatments were regressed against their M7 value. The regression was then used to calculate the expected yield at a preindustrial M7 of 10 ppb.

The calculations for obtaining the observed RY for the experimental data assume that the response of yield to increasing $O_3$
concentrations is approximately linear, which is verified in the literature (Pleijel et al., 2022). However, the effect of $O_3$ on leaf and grain protein, and grain amino acids has received far less attention in the literature, and it is unknown if their response to increasing $O_3$ is also linear. These factors meant it was not possible to estimate preindustrial leaf and grain protein and grain AA concentrations. Instead, we focus on the reduction in leaf and grain protein, and grain AAs under the elevated, as compared to the ambient, $O_3$ treatment.

## 3. Parameterisation and calibration of DO₃SE-CropN model

### 3.1 Experimental datasets

Datasets for training and evaluation of the DO₃SE-CropN model were taken from three years of field experiments for wheat harvested between March 2016-2018 at the Botanical Garden, Banaras Hindu University, Varanasi, India using the HUW234 and HD3118 cultivars. The cultivars are both late sown and heat tolerant wheat varieties. For all years, O₃ fumigation began 3 days after seed germination, the 13th, 14th and 15th of December respectively for the 2015, 2016 and 2017 sowing on the 14th of December. The wheat was exposed to ambient O₃ concentrations and an elevated O₃ treatment (ambient + 20 ppb), with the seasonal maximum O₃ concentrations ranging from 80-100 ppb, and an average ambient M7 of 48 ppb across 2017 and 2018. Across all experiments the wheat was sown on the 5th of December and harvested on the 30th of March. The wheat was grown in non-filtered open top chambers across all three years. The wheat did not experience any soil water or N stress. For greater detail of the experimental set up and measurements taken, the reader is referred to Yadav et al. (2020) and Yadav et al. (2021). A scaling factor was applied to each AA concentration in Yadav et al. (2020) based on the mean concentration of AA in Siddiqi et al. (2020) to ensure values were consistent with the wider literature on AA concentrations for Indian wheat.

The meteorological data for the model input was taken from an onsite weather and O₃ monitoring system. The input temperature data were corrected for the heating effect of the open top chambers, with the chambers found to be approximately 2°C warmer than the ambient air (see supplementary information). Due to gaps in the hourly meteorological data, gap filling was performed according to Emberson et al. (2021).

### 3.2 Model calibration and evaluation

### 3.2.1 Model calibration

The calibration for DO₃SE-CropN is performed sequentially to allow the interactions between parameters at each stage to be limited (Cook et al., 2024). The key parameters calibrated in the DO₃SE-CropN model are given in Cook et al. (2024) and the same method of calibration is used in this study. In the present study, there are 3 additional parameters to calibrate based on the newly introduced antioxidant module: $fst_{end}$, $a_{leaf}$ and, $a_{stem}$.

The maximum catalytic rate at 25°C ($V_{cmax,25}$) and the maximum rate of electron transport at 25°C ($J_{max,25}$) were fixed at the values provided by Yadav et al. (2020) in their supplementary data. The author's supplementary data on maximum photosynthetic rate were combined with data provided by the authors on maximum stomatal conductance to vary the species-specific sensitivity of stomatal conductance to assimilation rate ($m$), and the parameter describing variation in stomatal conductance in response to VPD ($VPD_0$), until a close match between photosynthetic rate and stomatal conductance was achieved (Yadav et al., 2020). Additional data provided by the authors of Yadav et al. (2020) were utilised to calculate the dark respiration rate; allowing calibration of the dark respiration coefficient for all simulations. Subsequently, the parameters controlling biomass accumulation and O₃ damage were calibrated using biomass data provided and assuming a seasonal

maximum LAI of 5. The $O_3$ damage parameters were incorporated at this stage due to high ambient $O_3$ concentrations which caused an $O_3$ induced reduction in yield even under the ambient treatment. The parameters controlling leaf and stem N were varied to achieve a close match for the leaf and grain protein simulations, as no stem N data was available for calibration. During this stage of the calibration, the gradient of the equations describing the effect of $O_3$ on N remobilisation of the leaf and stem were set to 0 to allow the newly developed antioxidant processes to be tested, as it was hypothesised in Cook et al. (2024) that the $O_3$ impact on N remobilisation occurs due to antioxidant processes. However, as the calibration was performed, the best results were achieved when the new antioxidant processes were used in combination with the previously developed $O_3$ effect on remobilisation. Therefore, the parameters controlling N remobilisation from the leaf and stem (calibrated in Cook et al. (2024)) were varied as little as possible from their defaults to allow the newly developed antioxidant processes to be parameterised. For a tabulation of parameters calibrated for, and the values they were calibrated to, please refer to the supplementary data.

Model parameters were calibrated using a combination of genetic algorithm and a trial-and-error approach, to minimise the difference between simulated and observed values while also retaining parameterisations that are physiologically realistic for the plant. For further details of the calibration method see Cook et al. (2024).

### 3.2.2 Model evaluation

The input data available for the present study were limited. Initially, the data were split in half with the 2017 data used for model calibration and the 2018 data used for the model evaluation. However, when looking at the results of the evaluation it was clear that the limited input data led to overfitting of the 2017 dataset (see supplementary Fig.'s S8 and S9). Therefore, to focus on the development of the modelling framework, all available data were used for model calibration. The root mean square error (RMSE) and $R^2$ were used to evaluate the model's suitability at simulating the yields and protein contents of the two cultivars using Scikit-Learn (Pedregosa et al., 2011). Using the $R^2$ calculation from Pedregosa et al. (2011) can give negative $R^2$ values, where a negative value means that using the mean of the observed values is a better fit to the data than using the model. In this paper the units of the RMSE are the same as the units of the model variable, e.g. for yield RMSE is reported in in $g\,m^{-2}$ and for protein percentage (% or $100gProtein/gDM^{-1}$) RMSE is reported as percentage points.

## 4. Results

### 4.1 Biomass and protein simulations

Overall, the calibrations for grain yield, and leaf and grain protein, were reasonable for both cultivars. The grain yield and RY loss simulations performed better for 2018 than 2017. However, there was little difference in the model's capacity to capture the leaf and grain protein concentrations, and the relative loss in these, under $O_3$ exposure between the years. From Fig. 3a, the grain yield calibration was satisfactory with a RMSE of 141 $gm^{-2}$, however it is clear that the calibration was able to simulate the grain DM better for 2018 than 2017. The underestimation of the grain DM in the 2017 dataset ranged between

35-46%. Further, the negative $R^2$ implies that using the mean of the observed data would be a better estimate of grain DM than the model (Pedregosa et al., 2011). The RY loss was captured much better than the grain DM. In Fig. 3b, the model captures the RY loss of cultivar HD3118 well. However, cultivar HUW234 has a large difference in RY loss between the two

335   years which the model was unable to capture. The average deviation of RY loss from the observed value is 2.5 percentage points excluding the HUW234 cultivar for 2018. When this cultivar is included, the deviation increases to 7 percentage points.

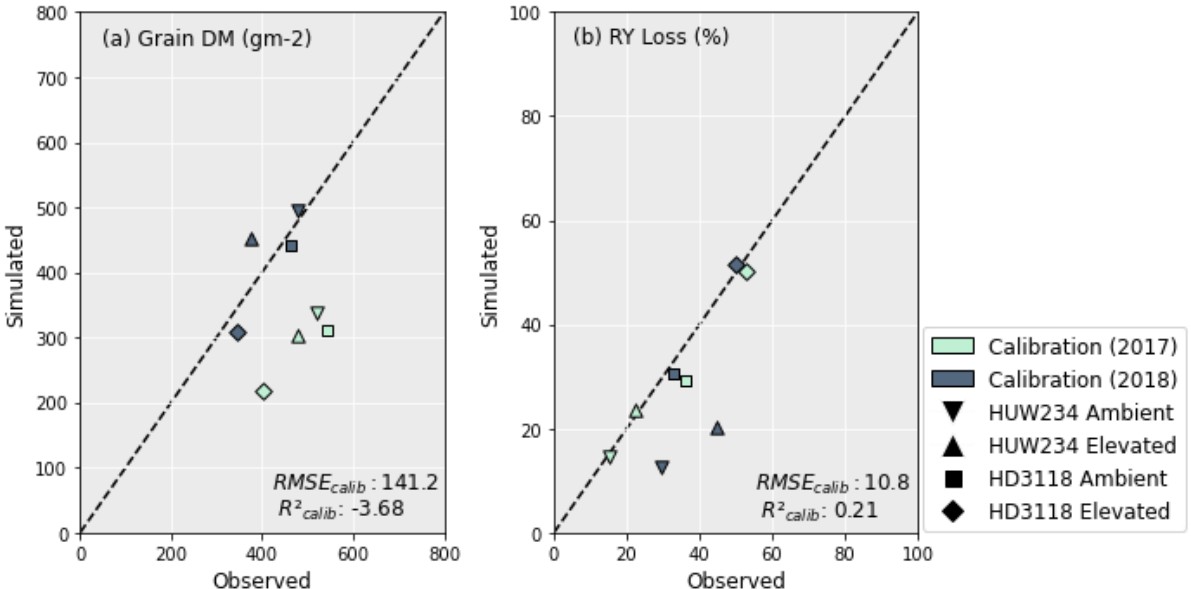

**Fig. 3: Calibration of grain DM (a) and RY loss (b) using the DO3SE-Crop model for the Varanasi dataset. RY loss was calculated comparative to preindustrial O3 concentrations(see section 2.5).**

Fig. 4 shows the grain and leaf protein simulations, and the relative protein (RP) loss between the ambient and elevated treatments. The grain protein (Fig. 4a) is captured better for 2017 than 2018, but overall, the results are good, with an $R^2$ of 0.5 and a RMSE of only 1.3%. The grain RP loss between the ambient and elevated O3 treatment is slightly overestimated for the HUW234 cultivar, and for the HD3118 cultivar in 2017 it is slightly underestimated, all by ~2.5 percentage points. However, in 2018 the grain RP loss of the HD3118 cultivar was heavily overestimated by ~6.5 percentage points.

The simulations of leaf protein (Fig. 4c) showed a good fit to the experimental data and were closer to the observed values than grain protein simulations, with an $R^2$ of 0.6 and a RMSE of 0.8%. Nevertheless, the model captured the pattern of the grain protein concentrations under ambient and elevated O3 concentrations better than the pattern of the leaf protein concentrations (Fig.'s 4a and 4c). The leaf RP loss (Fig. 4d) was not well captured. For the HD3118 cultivar, the leaf RP loss was underestimated, and for the HUW234 cultivar it was overestimated. For the HUW234 cultivar leaf RP loss was

overestimated by ~42 percentage points and for the HD3118 cultivar, the RP losses were more variable, with the leaf RP loss underestimated by ~13.5 and ~25 percentage points for 2017 and 2018 respectively.

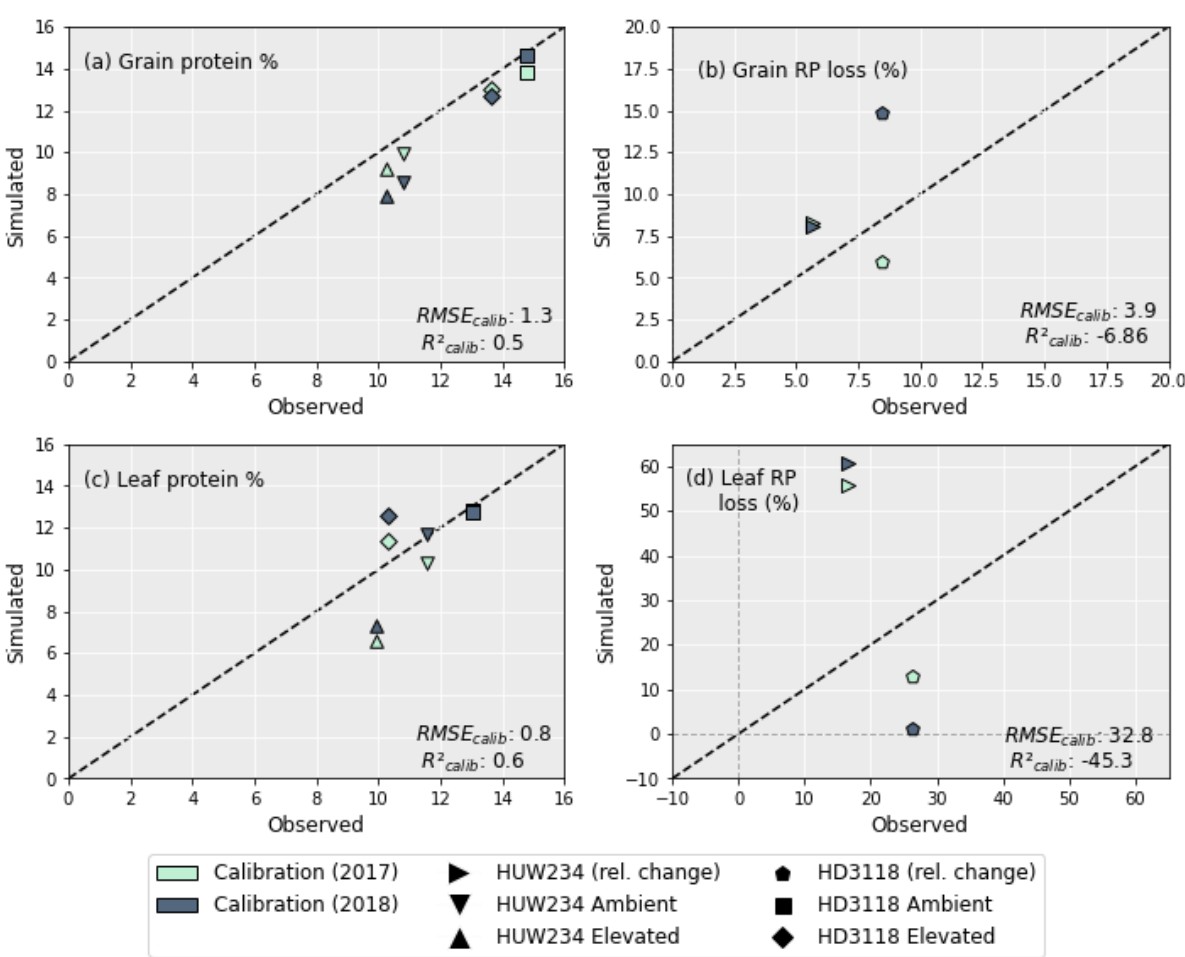

**Fig. 4: The concentration of grain (a) and leaf (c) protein of HUW234 and HD3118 cultivars under ambient and elevated O₃. Calibration of the RP loss in grain (b) and leaf (d). In figure (b) the relative change in grain protein for the HUW234 cultivar for the years 2017 and 2018 was almost identical, hence the overlaid points. In figure (c) the leaf protein concentration for HD3118 cultivar in the ambient treatment was almost identical for 2017 and 2018, giving the overlaid points. The RMSE and R² of the calibration are indicated on the plot.**

## 4.2 AA simulations

Lysine and methionine were the key AAs focussed on as they were found to be the most limiting to protein production under O₃ exposure (Yadav et al., 2020). To calculate their concentrations the grain protein concentrations (Fig. 4) were used along with the regressions from Liu et al. (2019). Fig.'s 5a and 5c show the concentration of methionine in the wheat grains is predicted better than the lysine concentrations, with a higher $R^2$ of 0.73 (compared with 0.31) and a lower RMSE (0.02 compared with 0.06). However, the decrease in both AA concentrations under O₃ exposure was not captured as well (Fig.'s 5b and 5d). For both lysine and methionine, the decrease in AA under O₃ exposure was heavily underestimated. The decrease in methionine for HUW234 and HD3118 was underestimated by 9 and 10.5 percentage points respectively. The decrease in

lysine concentrations were underestimated by 37 and 19 percentage points for HUW234 and HD3118 respectively. The decrease in AA concentration for HUW234 was similar between years for both methionine and lysine, whereas for the HD3118 cultivar, the simulations showed a drastically different decrease in AA concentration between years.

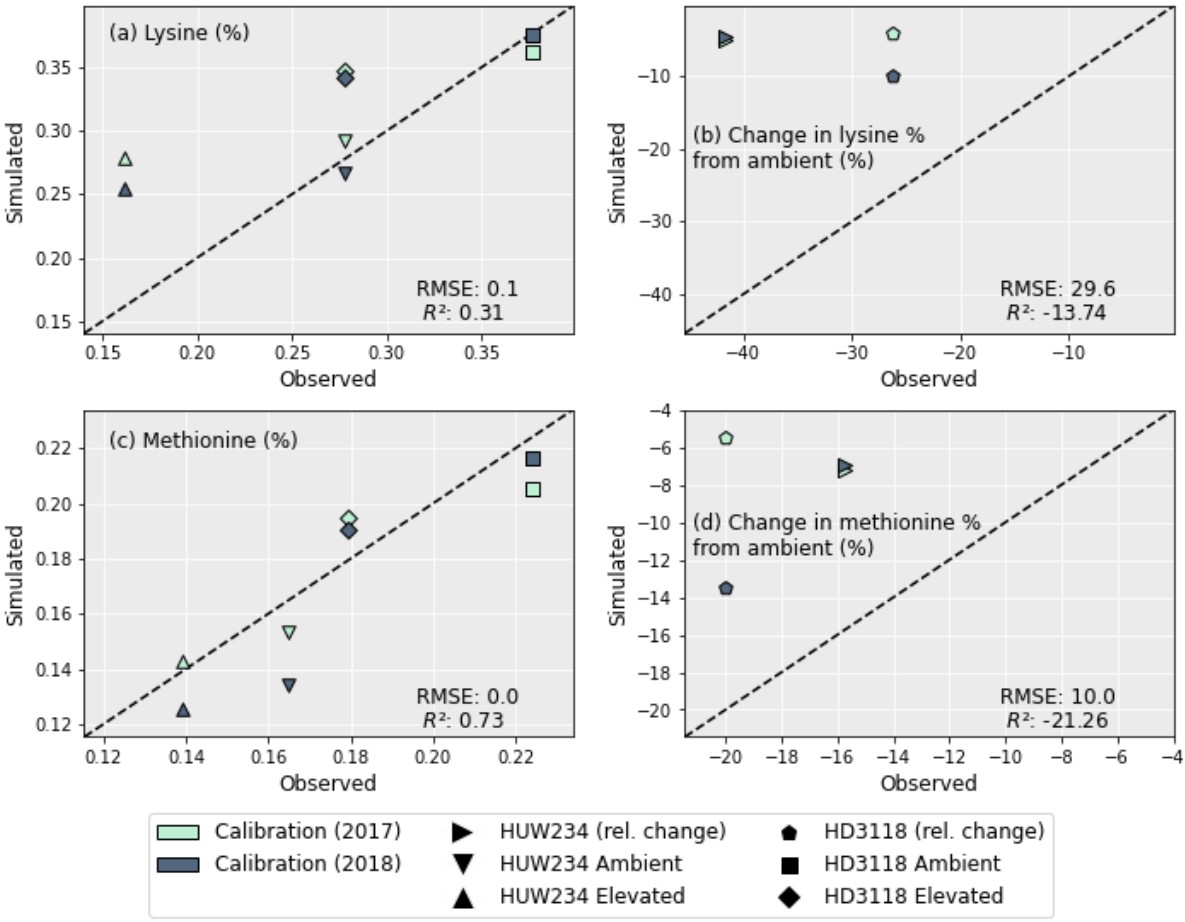

 **Fig. 5: A comparison of the simulated concentrations of lysine (a) and methionine (c) for the different cultivars and years of treatment. The change in AA from ambient for lysine (b) and methionine (d) is also shown for both cultivars.**

**4.3 DIAAS of the nutritionally relevant AA**

Methionine and lysine are the most limiting AA for protein production for the HUW234 and HD3118 cultivars, and experience a decrease in concentration under $O_3$ exposure (Yadav et al., 2020). Since the relative impact of $O_3$ on the AA concentrations was not captured well, the observed concentrations of the AA were used to calculate the DIAAS score, with the value that would be obtained if using the simulated outputs in brackets. After calculating the reference ratios for lysine and methionine using Eq. (3), lysine was found to give the lowest reference ratio for all $O_3$ treatments and cultivars, and was used to calculate the DIAAS using Eq. (4). Table 1 shows the results of the DIAAS calculation. Using the observed data, both cultivars experience a decrease in protein quality under elevated $O_3$ with the HUW234 cultivar experiencing a greater reduction than

the HD3118 cultivar. Overall, the quality of wheat protein was lower for children aged between 6 months – 3 years than for older children and adults (>3 years). When using the simulated outputs to calculate the DIAAS, there is an increase in protein quality under $O_3$ exposure. This discrepancy occurs due to the construct of the DIAAS equation. As the decrease in AA concentrations under $O_3$ were underestimated by $DO_3SE$-CropN in comparison to the grain protein (see Fig.'s 4 and 5), it led to a greater ratio of grain AA to grain protein (Eq. (3)). The greater ratio under elevated $O_3$ then led to a higher value DIAAS under the treatment compared to the ambient, though this would not be the case in reality.

Table 1: The DIAAS for the HUW234, and HD3118 cultivars under the two $O_3$ treatments and for the age categories 6 months – 3 years, and for older children and adults (>3 years). The reduction in DIAAS under $O_3$ for the HUW234 and HD3118 cultivars was also calculated. The numbers in brackets represent the DIAAS score calculated using model outputs, the average AA and protein concentrations across the 2017 and 2018 simulations were used in the calculation.

| | DIAAS | | | | | |
|---|---|---|---|---|---|---|
| Age category | HUW234 Ambient | HUW234 Elevated | HUW234 rel. change | HD3118 Ambient | HD3118 Elevated | HD3118 rel. change |
| DIAAS Score (> 3 years) | 49.9 (59.1) | 31.0 (61.1) | - 37.9% (+3.3%) | 49.9 (50.8) | 39.9 (52.4) | - 20.1% (+3.2%) |
| DIAAS Score (6 months - 3 years) | 42.0 (49.80) | 26.1 (51.5) | - 37.9% (+3.3%) | 42.0 (42.8) | 33.6 (44.1) | - 20.1% (+3.2%) |

## 4.4 Difference between the 2017 and 2018 simulations

After performing the simulations for 2017 and 2018 in section 4.1, it was clear there was a large difference in grain DM for the two years. The reasons for this discrepancy are important to understand since uncertainties on grain DM simulation will compound errors in protein concentration and yield (Cook et al., 2024). To investigate the grain DM discrepancy further, the meteorological variables, stomatal conductance and photosynthetic rate were plotted for both years. The accumulation of biomass each day and the LAI were overlaid to see if there were any differences that could explain the large difference in biomass. The temperature in 2018 was greater at the beginning and end of the growing season compared to 2017 (Fig. 6). The reverse was true for relative humidity (Fig. S2). With relation to the other inputs, air pressure, precipitation and wind speed had negligible differences between the years (Fig.'s S1, S3 and S4). $O_3$ concentrations were generally greater in 2017 than 2018 (Fig. 7 and Fig. S11), and photosynthetic photon flux density (PPFD) was greater at the start of the growing season in 2018 (Fig. 8). Daily photosynthetic rate was mostly greater in 2018 than 2017 and showed the same pattern for both cultivars (Fig.'s 9 and S5). The difference in stomatal conductance between the years for both cultivars mimicked the shape of the photosynthetic rate plots (Fig.'s S6 and S7). Given that the $O_3$ effect is more strongly determined by senescence than the instantaneous impact on photosynthesis (Pande et al., 2024b), and senescence onset did not differ strongly between years (Fig. S11), it is unlikely that the differences in yield were caused by $O_3$ effects. Instead, it is likely that 2018's higher early-season PPFD and temperatures, along with lower RH, promoted earlier LAI development and increased biomass production in simulations.

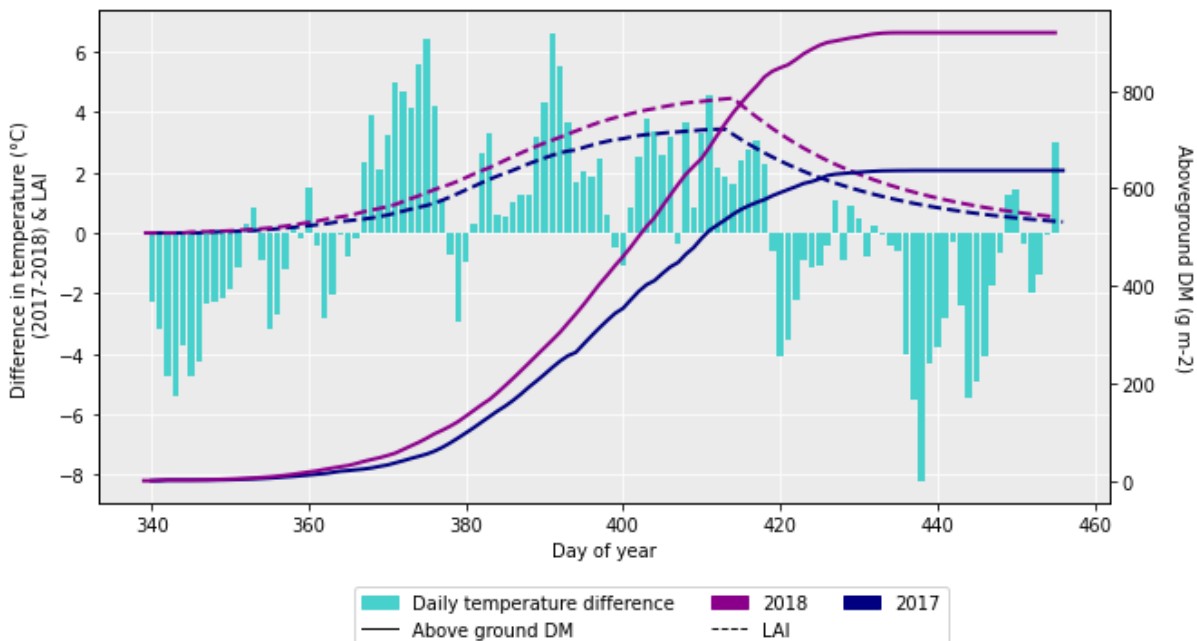

**Fig. 6: Plot of the difference in daily temperature between 2017 and 2018 (where 2018's temperatures were subtracted from 2017's temperatures), along with the difference in aboveground DM accumulation for the ambient treatment for both years and the LAI profiles. The LAI and aboveground DM profiles are for the HUW234 cultivar.**

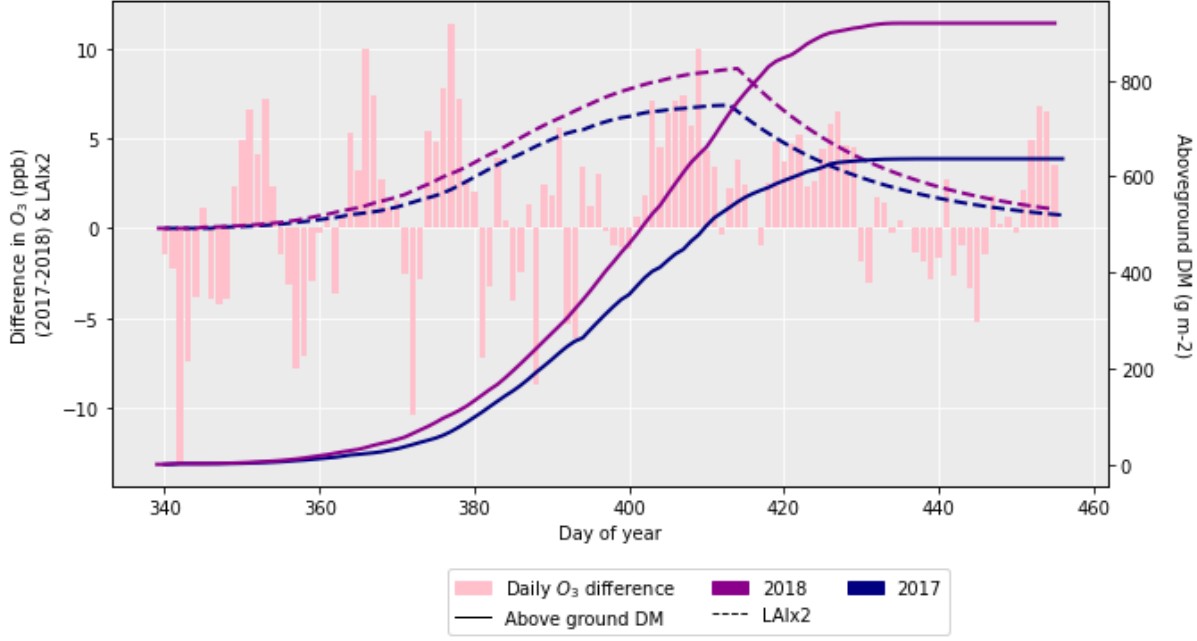

**Fig. 7: The difference in daily $O_3$ between 2017 and 2018 (where the $O_3$ concentrations for 2018 were subtracted from those for 2017) along with the difference in aboveground DM accumulation for the ambient treatment for both years and the LAI profiles. LAI has been multiplied by 2 to easier show the profile. The LAI and aboveground DM profiles are for the HUW234 cultivar.**

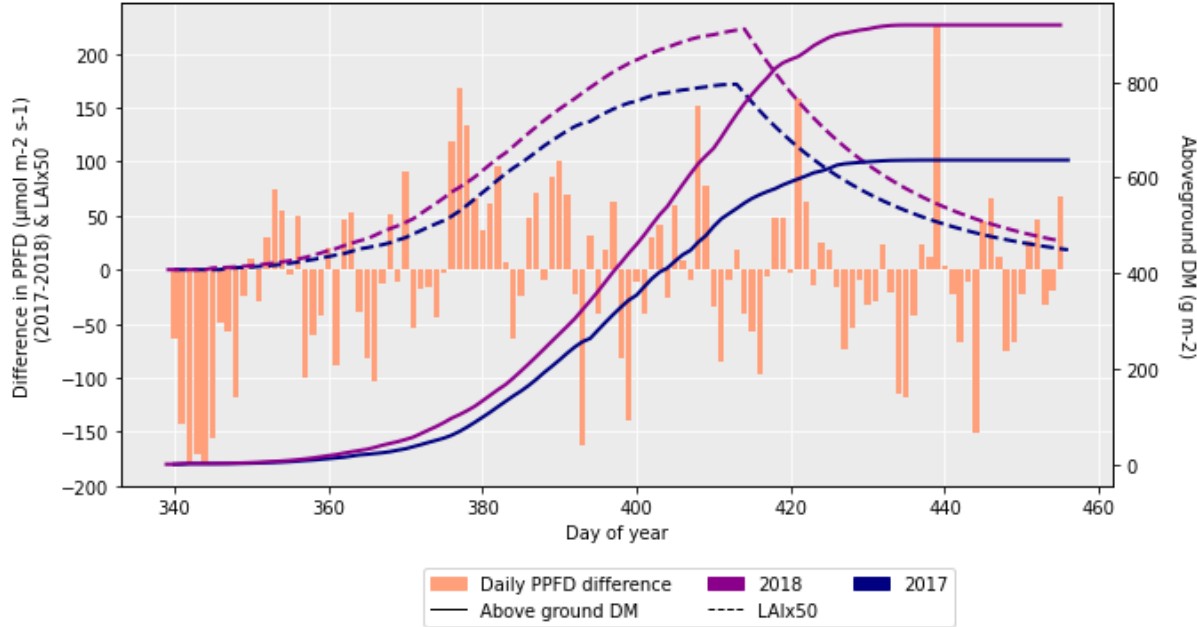

**Fig. 8: The difference in daily PPFD between 2017 and 2018 (where the PPFD for 2018 was subtracted from that for 2017) along with the difference in aboveground DM accumulation for the ambient treatment for both years and the LAI profiles. LAI has been multiplied by 50 to more clearly show the profile. The LAI and aboveground DM profiles are for the HUW234 cultivar.**

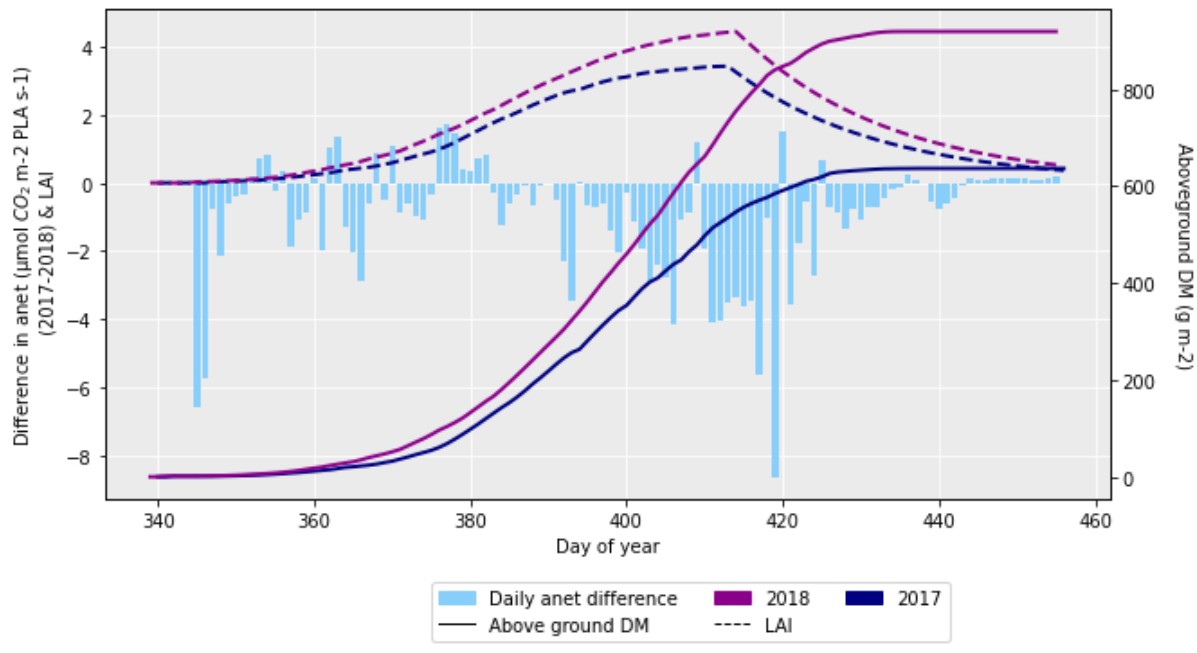

**Fig. 9: The difference in net photosynthetic rate for 2017 and 2018 (where the net photosynthetic rate for 2018 was subtracted from that for 2017) along with the difference in aboveground DM accumulation and LAI for the ambient treatment for the 2 years. The LAI and aboveground DM profiles are for the HUW234 cultivar.**

## 5 Discussion

We developed the DO$_3$SE-CropN model to address a current limitation in the ability of crop models to assess the effects of O$_3$ stress on not only crop yields, but also quality. We describe the further development and applications of the model to simulate O$_3$ effects on nutritionally important AA based on current understanding of antioxidant processes and implications for N remobilisation. This work is important since currently there are few models that consider protein quality in their simulations. CN-Wheat considers AA from the perspective of being used for leaf, stem and grain protein production (Barillot et al., 2016). They do not explicitly consider the AA of the wheat grains, relevant for nutrition (Barillot et al., 2016). In SiriusQuality, Martre et al. (2006) consider the fractions of N that are split between gliadin, glutenin, albumin-globulin and other proteins in the wheat grains as a measure of wheat quality for bread production, not human nutrition. To our current knowledge, Liu et al. (2019) are the only authors who have extended a crop model to simulate protein quality, in the form of AA, from the perspective of human nutrition. In their work they extended the CERES-Wheat model to simulate lysine and other indispensable AA concentrations. Further, none of these crop models that consider crop quality have incorporated the effects of antioxidants. In our study we extend the equations used by Liu et al. (2019) to produce the first framework by which the effect of O$_3$ on protein quality (through antioxidants, AA and the DIAAS) can be captured.

### 5.1 Ability of the model to simulate DM and protein

The present model was able to reproduce the observed grain DM for 2018 but underestimated it for 2017 due to differences in meteorology that triggered earlier LAI development, leading to greater photosynthesis and biomass production in the model for 2018. The model was able to capture the RY loss of the HD3118 cultivar well for both years. However, the HUW234 cultivar experienced a large difference in RY loss between the two years, with the model only able to capture the RY loss well for one year. With only 2 years of data, it was not possible to determine which of the observed RY loss is the most common response for HUW234. Data for additional O$_3$ treatments and years are required to develop a more robust model parameterisation for different meteorological conditionsand cultivars.

While the model underestimated the grain DM for 2017 by ~ 40%, there appears to have been no effect of this underestimation on the capacity of the model to capture grain protein concentration (100gProtein gDM$^{-1}$). A possible reason for this is that the lower photosynthesis in 2017, led to lower simulations of leaf and stem biomass. As a result, the N required by the leaf and stem for growth in the model was reduced, leading to lower N accumulation in these parts. Upon remobilisation to the grains, the reduction in N accumulated by the grains in 2017 compared to 2018, along with the reduced grain DM in 2017, led to similar protein concentrations. The model's ability to reproduce the observed grain protein concentration, despite yield discrepancies, suggests that the underlying N allocation and remobilisation equations reasonably approximate plant processes. This outcome supports the reliability of the equations, though further validation is needed to confirm their accuracy. Future work should focus on improving the model's estimates of protein yield and concentration so that O$_3$ threats to food security can be assessed with greater confidence.

In the model, there was a strong interdependence between the parameters controlling protein accumulation in the leaf and stem with grain protein, which is to be expected as protein remobilisation from the leaf and stem are key contributors to grain protein (Feller and Fischer, 1994; Gaju et al., 2014; Nehe et al., 2020). On calibrating the model, this interdependence meant that any attempt to improve the model's accuracy in capturing the decrease in leaf protein under elevated $O_3$ resulted in a reduced model accuracy in capturing the decrease in grain protein under elevated $O_3$, and vice-versa (see Fig.'s 4b and 4d). This meant that there was a trade-off between calibrating leaf and grain RP loss under $O_3$ exposure. No data was available on stem RP loss, so the accuracy of the stem parameterisation is unclear. Given this study focussed on grain quality, capturing the grain RP loss under $O_3$ was prioritised over the leaf. If the model is not able to match the decrease in leaf protein with the corresponding decrease in grain protein under $O_3$ exposure for a given cultivar, then it implies a problem with the parameterisation or model construct. Regarding the parameterisation, leaf DM data were not available which would affect leaf N, and hence protein, accumulation. Therefore, in the future leaf (and stem) DM data at anthesis and harvest would aid in parameterising the equations describing the partitioning of photosynthate each day and could improve simulations of RP loss.

## 5.2 Ability of the model to simulate AA concentrations

To date, there is only one study (by Yadav et al. (2020)) that has investigated the effect of elevated $O_3$ on the AA concentrations of wheat . Data from this study was used to calibrate and evaluate the DO$_3$SE-CropN model, as well as test the framework for the AA simulations. While the grain methionine concentrations were reproduced well, the grain lysine concentrations were overestimated for the elevated $O_3$ treatment. It is also clear to see that the reduction in concentrations of both lysine and methionine was underestimated by the DO$_3$SE-CropN model. The AA concentrations were calculated using regressions linking protein concentrations to AAs from Liu et al. (2019), which were constructed using data from 48 field experiments from major wheat producing areas in China. Approximately 95% of wheat grown in China is winter wheat (United States Department of Agriculture, 2022), and most of the cultivars used to produce the regressions were winter wheat (Liu et al., 2019). However, the model was parameterised for Indian spring wheat. Given the differences between the growing conditions in India and China, and spring and winter wheat, deviations in simulations of lysine and methionine concentrations from the observed are to be expected. Additionally, Liu et al. (2019) did not include experiments with differing levels of $O_3$ in their treatments. For lysine, this has culminated in a much better simulation of the AA concentrations under ambient $O_3$ compared to the elevated treatment. For both lysine and methionine, using the regressions alone to convert grain protein to grain AA concentrations was not sufficient to account for the $O_3$ effect on grain quality. Additionally, there is currently a knowledge gap (discussed further in section 5.3) relating to our understanding of $O_3$'s effects on both antioxidants and grain quality, which affects not only the construction of the model but also its parameterisation. Suggestions for more specific experiments which could reduce the knowledge gap for both modelling and understanding the effect of under $O_3$ exposure on grain protein and AAs are discussed in sections 5.3, 5.4 and 5.7. Nevertheless, it is clear that additional experimental data on $O_3$'s effect on grain AA would be beneficial for not only model development, but improving confidence for modelling results.

## 5.3 Modelling antioxidant processes under $O_3$ exposure

The first iteration of the $DO_3SE$-CropN model simulated the decrease in grain protein yield (gProtein m$^{-2}$), and increase in grain protein concentration (100gProtein gDM$^{-1}$) experienced by European and Chinese wheat cultivars (Broberg et al., 2015; Cook et al., 2024). However, Indian wheat has been shown to experience a decrease in both protein yield and concentration under $O_3$ exposure (Mishra et al., 2013; Yadav et al., 2020). Through the incorporation of antioxidant processes, the present model is now able to capture the decrease in protein concentration, and yield, of protein in Indian wheat under $O_3$ exposure, improving the regional applicability and nutritional relevance of the model.

The design of the antioxidant equations has several benefits which make it useful for further applications. Firstly, the structure of Eq. 1 means that it could be translated to other stressors provided they have a similar mechanism of damage to $O_3$, meaning the framework is flexible. Drought and high temperature stress are good candidates for this framework as they are ROS mediated, like $O_3$, and cause a reduction in both grain yield and protein content (Broberg et al., 2015, 2023; Mariem et al., 2021). The effect of heat stress on antioxidant production, and hence grain quality, could be incorporated by modifying Eq. 1 and Fig. 2 to incorporate the duration (and potentially timing) of the stress as these are the key factors affecting grain yield under heat stress (Balla et al., 2019). For drought stress, the duration of the stress would be useful, but there would need to be an additional effect of drought on reducing nutrient uptake (as this affects grain quality) (Faisal et al., 2017; Rijal et al., 2020). The second benefit of the framework is that it is simple. It does not require a large number of additional parameters, which reduces the complexity of the modelling process and makes it easier for other modellers to introduce into their models. Thirdly, the framework is compatible with the structure of other models that simulate plant N. The equations can be used to simply divide leaf and stem N into pools that are accessible or inaccessible (antioxidants) to the grain. Following this, the modeller only needs to ensure that any N remobilised from the leaf and stem to the grain comes from the accessible pool.

It was hypothesised that the introduction of the antioxidant processes would replace the previous $O_3$ effect on leaf and stem residual N that was parameterised in Cook et al. (2024), as it was previously hypothesised that the increase in residual N occurred as a result of antioxidant production (Brewster et al., 2024; Cook et al., 2024; Sarkar et al., 2010). However, during model calibration it was noted that the simulations of leaf and grain protein were improved when both processes were used in combination (see model parameterisation in supplementary information). There are two potential explanations for this: 1) The shape of the antioxidant response to $O_3$ is such that the two effects working in combination are a more effective approximation, meaning further data to investigate the effect could provide insight into the truer shape of the response, 2) $O_3$ has an effect on N remobilisation from the leaf and stem to the grains that is separate to antioxidant production. For example, ROS have been shown to oxidise proteins which would decrease protein concentrations but lead to greater residual N in the leaf and stem (Gill and Tuteja, 2010). Given this, and the previously described trade-off when calibrating leaf and grain RP loss, there is clearly a knowledge gap in our current understanding of antioxidant production and remobilisation of nutrients under $O_3$ exposure. Therefore, a study with multiple $O_3$ treatments that identifies the proportion of N in the leaf and stem at anthesis, and the leaf, stem and grains at harvest, as well as the corresponding proportion of proteins, would allow identification of how much N is

associated with proteins, and whether this fraction changes under $O_3$ exposure and affects N remobilisation. Such data would also allow further development of the antioxidant equations in this study, as for simplicity, and lack of data to test a more complex relationship, we have assumed linearity, but this may not be the case. Additionally, identification of the N associated with antioxidants at anthesis and harvest, and how these change under $O_3$ exposure, would also allow further development of the antioxidant equations. If combined with protein measurements at anthesis and harvest, mechanistic understanding of $O_3$ impacts on protein, antioxidant processes and grain filling with N could be developed further and used to refine existing model processes.

## 5.4 Antioxidant processes and grain quality

For consideration of $O_3$ effects on nutrition, it is important to consider the protein quality, in addition to its concentration. From a dietary perspective, indispensable AA, such as lysine and methionine, are the most important to consider when thinking about protein quality as they cannot be produced by the body and must be obtained through diet (Elango et al., 2008; Shewry and Hey, 2015). Lysine and methionine are key as they are the AA available in the lowest quantity in wheat exposed to $O_3$, and therefore limit the body's capacity to produce proteins from them (Yadav et al., 2020). If a person does not consume enough, or a high enough quality, of protein, then they are at risk of wasting and loss of muscle function (Medek et al., 2017). Understanding how $O_3$ induced changes to wheat protein will affect protein quality, and hence quality of diet, is key in understanding $O_3$ effects on human nutrition and its potential role exacerbating malnutrition.

The regressions from Liu et al. (2019) were used to simulate grain lysine and methionine concentrations as these were the most limiting for protein production under $O_3$ exposure (Yadav et al., 2020). However, there is variability in the response of AA in wheat grains under $O_3$ due to the differential activation of metabolic pathways under stress (Ali et al., 2019; G A et al., 2024; Li et al., 2024; Wang et al., 2018). Yadav et al. (2020) found that while overall protein concentrations decreased under elevated $O_3$, lysine and methionine concentrations decreased, while grain serine concentrations increased. The responses also differed between cultivars with HUW234 having an increase in threonine, while HD3118 had a decrease (Yadav et al., 2020). During stress conditions, the concentrations of AA vary to enhance plant defence mechanisms against abiotic stressors (Ali et al., 2019; G A et al., 2024; Li et al., 2024; Wang et al., 2018). In HUW234 and HD3118, lysine concentrations decreased under elevated $O_3$, due to its breakdown for energy production and plant defence (Ali et al., 2019; Yadav et al., 2020). Lysine breakdown produces proline, the concentration of which increased in both cultivars, which has been shown to protect against ROS-induced oxidative damage (Nayyar and Walia, 2003; Yadav et al., 2020; Yang et al., 2020). Additionally, the concentration of methionine decreased in both cultivars under elevated $O_3$ (Yadav et al., 2020). The decrease is likely due to methionine's role as an antioxidant, and that it is very sensitive to oxidation by ROS (Ali et al., 2019). The changes in AA aid in the maintenance of photosynthetic rate and protection of photosynthetic pigments from ROS (Kaur and Kapoor, 2021; Naidu et al., 1991; Simon-Sarkadi and Galiba, 1996). The specific response of an AA to abiotic stress is cultivar specific and depends on the intensity of the stress (Ali et al., 2019). As a result, grain AA concentrations are linked to the stress response of the plant under $O_3$. Measurements of AA concentrations under multiple $O_3$ treatments would help to elucidate the shape of the response

of AA's to $O_3$ stress. This is a field which has largely been neglected with only Yadav et al. (2020) having investigated it so far. Such data would allow the effect of $O_3$ on nutrition to be better understood.

## 5.5 Protein quality estimates using the DIAAS

Through extending DO$_3$SE-CropN to simulate the DIAAS, estimates of protein quality are translated into a metric that is commonly used to assess dietary quality in the nutrition field (e.g. Kurpad and Thomas (2020)). Using the observed data, the HUW234 cultivar experienced the greatest loss in protein quality under increased $O_3$ concentrations, despite showing the smallest RP and RY loss. The reason for this is that HUW234 experienced the greatest decrease in lysine concentrations, and lysine is the most limiting AA in wheat (Meybodi et al., 2019; Siddiqi et al., 2020). The DO$_3$SE-CropN model was not able to reproduce the reduction in protein quality calculated through the DIAAS as it was not able to reproduce the magnitude of the decrease in protein and lysine concentrations under elevated $O_3$ for either cultivar (Table 1, Fig.'s 5b and 5d). Using the observed data, the calculations of DIAAS were the same for both cultivars due to the scaling factor used for the AA (see section 3.1) but, in reality, the DIAAS would differ between the cultivars. While using the simulations of grain protein and AA's was able to produce a difference in DIAAS between cultivars, it was only able to reproduce the DIAAS calculated from the observed data for the HD3118 cultivar in the ambient $O_3$ treatment, as the protein and lysine concentrations were captured well for this cultivar and treatment. To develop crop models that use the DIAAS to understand the reduction in protein quality under abiotic stress, the reduction in grain protein and the most limiting AA's for protein production under that stress need to be understood.

## 5.6 Data requirements for effective model calibration

Initially in this study, the data were split in half, and the 2017 data were used for model calibration and the 2018 for evaluation. However, due to the model overfitting to the 2017 dataset (see supplementary data), the decision was made to utilise all available data for calibration. This allowed the paper to focus on the development of the antioxidant processes and protein quality simulations. Should future work utilise the antioxidant or protein quality framework presented in this work, a thorough model calibration and evaluation is recommended. Calibrations that use data from contrasting growing conditions, such as different growing seasons/years, sowing dates or experimental conditions have been shown to reduce the chance of multiple combinations of parameters giving the same answer (equifinality), reduce model uncertainty, and improve simulation accuracy (He et al., 2017; Zhang et al., 2023). This is likely a result of achieving a truer parametrisation for the cultivar, leading to improved generalisation of the model upon application (Wallach, 2011). Hence, if there are few growing seasons of data available, it would be helpful to have data spanning a range of crop treatments.

## 5.7 Further work for understanding $O_3$ effects on wheat nutrition

Current risk assessments of $O_3$'s effect on Indian wheat yields have predicted the greatest yield reductions across the IGP and eastern India due to the high $O_3$ concentrations in this region as well as meteorological conditions that favour plant $O_3$ uptake

(Droutsas, 2020; Mills et al., 2018a; Tai et al., 2021). (These estimates exclude concentration-response methods, which are not as biologically relevant, since these do not include the modifying effect of meteorology on $O_3$ uptake the spatial distribution of yield losses differs (Emberson et al., 2000; Pleijel et al., 2022).) From this, we can hypothesise that nutrition impacts will also be greater in these regions, though the specific response will vary by cultivar. However, the work of the present study does not just have applications for India. Understanding cultivar-specific responses to increasing $O_3$ concentrations will be important for food security globally in order to breed cultivars that can maintain yields and protein quantity, as well as quality, in the future. Additionally, it can be seen in the calculations of the DIAAS score, and reflected in the wider literature, that the quality of protein in wheat is low, even without the impact of $O_3$, which will exacerbate protein deficiencies in consumers who rely on wheat based diets (Swaminathan et al., 2012). Therefore, to reduce malnutrition, cultivars with a high protein quality, that can maintain yields and protein concentration under $O_3$ exposure should be investigated for their potential to maintain wheat supply and quality under conditions of elevated $O_3$ concentration. Additionally, existing barriers to diet diversification need to be overcome, so that individuals may have access to higher quality protein sources (Agrawal et al., 2019).

To develop an understanding of cultivar specific responses to abiotic stress, a modelling approach similar to that used in this study would be useful, as such a model can capture the effect of antioxidant processes under stress on grain quality. To ensure the applicability of the model in addressing this goal there are a few existing barriers identified in this study:

1) Before model application, models need to be thoroughly calibrated and evaluated. To perform a thorough calibration and evaluation, a range of treatments and/or years of data need to be available to provide a set of calibration parameters that are more general for that cultivar and prevent over-fitting. Additionally, obtaining leaf and stem DM anthesis and harvest will aid in parameterising partitioning and remobilisation of photosynthate.

2) Differences in meteorological conditions between the two years of experiments in the present study had a large effect on simulations of grain DM. The effect of meteorology on simulations of photosynthetic processes and biomass production in crop models should be further investigated in the future to ascertain crop model sensitivity to input data choices.

3) To advance the antioxidant equations, and understand $O_3$ effects on grain quality, an $O_3$ exposure (e.g. FACE, OTC or solardome) experiment measuring total N and protein content, and N and protein concentrations in the leaf and stem at anthesis, and harvest stages under varying $O_3$ treatments should be conducted. The proportion of N associated with specific antioxidants (such as glutathione and enzymatic antioxidants) under these $O_3$ treatments should also be obtained to improve mechanistic understanding of plant antioxidant response to $O_3$. This can be used to further develop the model, as it is anticipated that increased allocation of N to antioxidant production in leaves and stems under $O_3$ stress reduces the N available for remobilisation to grains during grain filling, leading to a decrease in grain protein concentration and altered amino acid profiles.

4) From the same $O_3$ exposure experiments, measurements of grain protein and AA concentrations for each $O_3$ treatment should be collected to produce relationships linking the two and how the relationship changes under the influence of $O_3$ to verify whether there is a trade-off between stress mitigation and nutritional quality. Such

relationships could be integrated in the model to improve its ability to simulate AA concentrations under stress, and hence provide more trustworthy estimates of protein quality.

Reliable estimates of DIAAS would allow dietary protein quality to be incorporated into $O_3$ risk assessments. Performing yield and nutrition-based risk assessments utilising AA and DIAAS simulations under future $O_3$ scenarios would allow for assessment of which wheat growing areas will experience a decrease in wheat protein quality as well as yield. Such results could then be combined with dietary surveys to evaluate adult's and children's risk of not getting enough, or a high enough quality of, food under increasing $O_3$.

**6. Conclusions**

In summary, the present study has developed a framework by which the antioxidant response of wheat under $O_3$ exposure can be incorporated into wheat quality simulations in the existing crop model DO$_3$SE-CropN. The key benefits of the framework are that it is flexible, simple and compatible with other crop models provided they simulate leaf and stem N. The AA's most limiting for human nutrition under $O_3$ exposure were found to be lysine and methionine. The new modelling framework

allowed the effect of high $O_3$ concentrations leading to a decrease in grain protein, lysine and methionine concentrations of Indian wheat to be simulated. Through calculations of the AA's, the FAO recommended metric for simulating wheat quality, the DIAAS, can be calculated. To improve the present model, we identified key experimental data needed to test and refine model formulations and parametrisations for a wider range of meteorological conditions and wheat cultivars. These include greater calibration data across multiple years and treatments with leaf and stem DM and N measurements, mechanistic

understanding of plant antioxidant response, and further development of relationships linking grain protein concentrations to AA concentrations under elevated $O_3$.

**Code availability**

An open version of the DO$_3$SE-Crop model, version 4.39.16 as used in the present study can be found at Bland (2024) and version 2.0 of the N module for DO$_3$SE-Crop is found at Cook (2024).

**Data availability**

Data from Yadav et al. (2020) and Yadav, Agrawal and Agrawal (2021) were used in the present study with additional data provided by Durgesh Singh Yadav (durgeshsinghy@gmail.com). Due to data ownership, please contact Durgesh Singh Yadav directly for access to required data.

## Author contributions

Conceptualisation: Jo Cook, Lisa Emberson, Felicity Hayes, Samarthia Thankappan, Durgesh Singh Yadav

Data curation: Jo Cook, Durgesh Singh Yadav, Nathan Booth

Formal analysis: Jo Cook

Methodology: Jo Cook, Lisa Emberson, Felicity Hayes, Samarthia Thankappan, Durgesh Singh Yadav

Software: Jo Cook (N module, antioxidant processes and $DO_3SE$-Crop), Sam Bland ($DO_3SE$-Crop), Pritha Pande ($DO_3SE$-Crop), Nathan Booth ($DO_3SE$-Crop), Lisa Emberson ($DO_3SE$-Crop)

Supervision: Lisa Emberson, Felicity Hayes, Samarthia Thankappan

Visualisation: Jo Cook

Writing – original draft preparation: Jo Cook

Writing – review and editing: Jo Cook, Lisa Emberson, Felicity Hayes, Samarthia Thankappan, Durgesh Singh Yadav, Nathan Booth, Pritha Pande, Sam Bland

## Funding

This work was supported by ACCE grant number NE/S00713X/1

## Competing interests

The authors declare that they have no conflict of interest.

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
