# Peer review of "Modelling the nutritional implications of ozone on wheat protein and amino acids"

_EGUsphere, 2024_

## Community Comment (CC1)

November 15, 2024

Comments by Owen R. Cooper (TOAR Scientific Coordinator of the Community Special Issue) on:

**Modelling the nutritional implications of ozone on wheat protein and amino acids**

Jo Cook, Durgesh Singh Yadav, Felicity Hayes, Nathan Booth, Sam Bland, Pritha Pande, Samarthia Thankappan, Lisa Emberson

EGUsphere [preprint], https://doi.org/10.5194/egusphere-2024-2968
Discussion started Oct. 10, 2024
Discussion closes Nov. 29, 2024

This review is by Owen Cooper, TOAR Scientific Coordinator of the TOAR-II Community Special Issue. I, or a member of the TOAR-II Steering Committee, will post comments on all papers submitted to the TOAR-II Community Special Issue, which is an inter-journal special issue accommodating submissions to six Copernicus journals:  ACP (lead journal), AMT, GMD, ESSD, ASCMO and BG. The primary purpose of these reviews is to identify any discrepancies across the TOAR-II submissions, and to allow the author teams time to address the discrepancies.  Additional comments may be included with the reviews. While O. Cooper and members of the TOAR Steering Committee may post open comments on papers submitted to the TOAR-II Community Special Issue, they are not involved with the decision to accept or reject a paper for publication, which is entirely handled by the journal's editorial team.

**Comments regarding TOAR-II guidelines:**

TOAR-II has produced two guidance documents to help authors develop their manuscripts so that results can be consistently compared across the wide range of studies that will be written for the TOAR-II Community Special Issue.  Both guidance documents can be found on the TOAR-II webpage: https://igacproject.org/activities/TOAR/TOAR-II

*The TOAR-II Community Special Issue Guidelines*:   In the spirit of collaboration and to allow TOAR-II findings to be directly comparable across publications, the TOAR-II Steering Committee has issued this set of guidelines regarding style, units, plotting scales, regional and tropospheric column comparisons, and tropopause definitions.

*The TOAR-II Recommendations for Statistical Analyses*:  The aim of this guidance note is to provide recommendations on best statistical practices and to ensure consistent communication of statistical analysis and associated uncertainty across TOAR publications. The scope includes approaches for reporting trends, a discussion of strengths and weaknesses of commonly used techniques, and calibrated language for the communication of uncertainty. Table 3 of the TOAR-II statistical guidelines provides calibrated language for describing trends and uncertainty, similar to the approach of IPCC, which allows trends to be discussed without having to use the problematic expression, "statistically significant".

**General comments:**

The authors have submitted an extremely well-written manuscript to the TOAR-II Community Special Issue, and the paper has already received two thorough reviews from the anonymous referees. I find the paper and its conclusions to be consistent with the findings of previous TOAR work and I only have a few minor comments, listed below.

Line 70-72
The first phase of TOAR was only able to find publicly available ozone data at 3 sites in India (Mills et al., 2018), and only two of these were in the wheat growing areas of northern India. Given that observations are so sparse, how confident are you that India has some of the highest ozone levels with regards to crops? Have additional ozone observations become available at more sites across India since the first phase of TOAR? Is your assessment based on models?

Line 69
Nitrous oxide should be nitrogen oxides

Line 244
29016 should be 2016

Line 410
Meteorology's doesn't seem to be the right work. Would "meteorological conditions" be more suitable?

---

## Author Response (AR1)

Dear Paul,

Please find below the point-by-point address of the reviewer and editor comments which we have addressed in the manuscript. Green text refers to our replies and the black text refers to the original reviewer comment. We also change the title of the manuscript from "Modelling the nutritional implications of ozone on wheat protein and amino acids" to "Modelling ozone-induced changes in wheat amino acids and protein quality using a process-based crop model". While the reviewers did not request a change to the title, after discussion with the team we feel this new title better represents the paper. We were concerned that including the term "nutritional implications" in the previous title may have led the reader to think we would talk about food access or availability, or human diet, when this is not the focus of the study.

**Response to RC1**

In this manuscript, Cook et al. present their study on **"Modelling the nutritional implications of ozone on wheat protein and amino acids"** to improve the DO$_3$SE-CropN model. Through simulations of nutrition-based ozone risk assessment in Indian wheat, the authors develop a flexible framework for crop models by incorporating the antioxidant responses, as well as leaf and stem nitrogen dynamics under ozone exposure. This study provides valuable data and is aligned with the journal's scope. The manuscript is well-written; however, some corrections are required to improve the quality.

I recommend publication after a relatively minor revision as follows:

Line 40: Update "Mills et al. 2018b" to "Mills et al. 2018a" for the first citation of Mills et al. in 2018. The subsequent citation with the same author and year should be referenced as "Mills et al. 2018b."

I thank the reviewer for bringing this to my attention. I contacted the copy-editing team for the journal, and they have informed me that the in-text references should match the bibliography which is sorted alphabetically. In my case, my first reference is to the tropospheric ozone assessment report paper from Mills et al. (2018), which occurs later alphabetically than the other Mills et al. (2018) paper and receives the tag "b" to distinguish it.

Mills, G. et al. (2018a). Ozone pollution will compromise efforts to increase global wheat production. Global Change Biology, 24 (8), pp.3560–3574. [Online]. Available at: doi:10.1111/gcb.14157.
Mills, G. et al. (2018b). Tropospheric Ozone Assessment Report: Present-day ozone distribution and trends relevant to human health. Elementa: Science of the Anthropocene, 6 (47). [Online]. Available at: doi:10.1525/elementa.302.

Based on the comments of the copy-editing team this reference does not require changing

Lines 40-42: Expand briefly on the TOAR-I report's findings regarding current ozone trends.

We thank the reviewer for their suggestion and have now included a summary of some of the key findings from the first phase of the tropospheric ozone assessment report:

"From the first phase of TOAR, it was observed that tropospheric O$_3$ increased globally in the 20th century, with atmospheric chemistry and climate modelling studies finding that O$_3$ production is greatest in mid to high latitudes due to greater emissions of O$_3$ pre-cursors

(Archibald et al., 2020; Cooper et al., 2014). Additionally, using the database Mills et al. (2018b) found that in East Asia $O_3$ concentration metrics for wheat growing locations where much greater than in Europe. Several authors for the first phase of TOAR commented on the underrepresentation of some key wheat producing areas (particularly India but also for China and Russia) in the database, which limited some of the analysis (Cooper et al., 2014; Mills et al., 2018; Schultz et al., 2017)."

Lines 42-44: Make this sentence clear by breaking it into two sentences.

We agree that the sentence was long and clunky, instead of breaking it into 2 we have removed the unnecessary information and simplified it to say: "This paper is part of the second phase of TOAR (https://igacproject.org/activities/TOAR/TOAR-II), which expands on the first phase to investigate $O_3$ impacts on human health and vegetation."

Lines 72-73: Provide possible reasons for the high $O_3$ concentration in this region for better context.

We thank the reviewer for this comment. We have added the following sentence to explain the high $O_3$ concentrations:

"These high $O_3$ burdens occur due to increasing pre-cursor emissions and insufficient pollution control measures (Archibald et al., 2020; Singh et al., 2023; Wang et al., 2023)."

Lines 76-77: Rephrase this sentence, as comparing present-day conditions to seasonal changes may not appropriately represent $O_3$ concentration variations.

We understand the point the reviewer makes here. We were aiming to highlight that the greatest increase in $O_3$ concentrations in India will occur during the wheat growing season, which will threaten wheat production further in the country. We appreciate this was not particularly clear. We provide further information on the findings of Kumar et al. (2018) which we then link to the dry, wheat growing season in the second sentence to make the $O_3$ concentration variations clearer.

"Using a Nested Regional Climate Model with Chemistry, Kumar et al. (2018) projected that $O_3$ concentrations across India will rise under RCP 8.5, while remaining comparable to current levels under RCP 6.0. For the dry, wheat growing season, the authors projected that $O_3$ concentrations across the IGP will increase under both RCP 6.0 and RCP 8.5, with a much larger increase under RCP 8.5."

Line 125: Introduce the existing $DO_3SE$-Crop model here, summarizing essential inputs and its application in a few sentences for reader comprehension.

We agree that it would be useful for the reader to understand this information in the introduction before reaching the methods section.

We add the following brief information to provide some context on the DO3SE-Crop model (purple text):

"Currently, only one model has been developed which captures the effect of $O_3$ on crop nutrition: $DO_3SE$-CropN (Cook et al., 2024). $DO_3SE$-CropN is built on the existing $DO_3SE$-Crop model, which takes inputs of hourly meteorology and $O_3$ concentrations to simulate crop phenology, $O_3$-impacted net photosynthesis, dry matter partitioning, grain filling and $O_3$ impacted crop senescence (Pande et al., 2024). The $DO_3SE$-CropN model then simulates crop

N, and models explicitly the effect of O$_3$ on reducing the amount of N from the leaves and stems that is available for the grain."

Line 156: "recovery from O$_3$ damage overnight" Please clarify the phrase.

We agree that the previous phrasing of the sentence is confusing. We modify it to be clear that it is the O$_3$ effect on photosynthesis that the plant can recover from overnight

"The DO$_3$SE-Crop model is a coupled stomatal conductance-photosynthesis model, which simulates stomatal O$_3$ uptake, its impact on photosynthesis which the plant can recover from overnight, as well as O$_3$ induced accelerated crop senescence (Pande et al., 2024)."

Line 172: Specify which antioxidants are considered for model integration, as "antioxidants" is a generalized term.

We thank the reviewer for their comment and understand that the current framing of this section makes the antioxidants considered unclear to the reader. In this case we do not consider specific antioxidants, we simply consider antioxidants as a general pool. We add the following make this clearer:

"For the purposes of this study, we do not consider individual antioxidants (e.g. superoxide dismutase (Tiwari and Agrawal, 2018)). Instead, we model a general pool of N that we hypothesise to be associated with antioxidants. This antioxidant pool of N is subsequently unavailable to the grain and is suggested to partially explain the decrease in grain protein of Indian wheat under O$_3$ exposure."

Line 347: "Error! Reference source not found. shows the results of the DIAAS calculation." Correct the sentence "Table 1 shows the results of the DIAAS calculation"

We thank the reviewer for pointing this out, we have now corrected this

Line 503: "(Error! Reference source not found., Fig.'s 5b and 5d)." Correct the sentence.

We thank the reviewer for pointing this out, this should have been Table 1 and we have now corrected this

Line 522: Replace "O3" with "O$_3$"

We have corrected this

Line 634: Reference is incomplete, please address the missing information.

We thank the reviewer for bringing this to our attention, this is a person's PhD thesis and so we have now included the associated university

We would like to extend our gratitude to the reviewer for their attention to detail and suggestions to the manuscript. We feel the clarity of the paper is much improved by their suggestions.

**Response to RC2**

This manuscript presents the development of the DO3SE-CropN model, which simulates reductions in protein and amino acids in wheat subjected to ozone (O$_3$) stress. The authors

effectively incorporate antioxidant processes, thereby increasing the model's applicability for predicting $O_3$-induced quality losses in crops. Including crucial amino acids, such as lysine and methionine, is essential for assessing the effects on protein quality. Further clarification regarding the specific enhancements to the original model framework and how these modifications rectify limitations in earlier versions would enhance the manuscript.

1. The authors illustrate the model's ability to predict protein quality, successfully simulating lysine and methionine concentrations in wheat grain. However, the underestimation of reductions in amino acid content induced by $O_3$, particularly for lysine, is significant. The manuscript should discuss potential reasons for these discrepancies, such as limitations in the underlying assumptions of antioxidant pathways.

We thank the reviewer for their suggestion. We agree that the discussion of AA simulations was limited. We have now incorporated an additional section into the discussion, in between previous sections 5.1 and 5.2, to better discuss the reasons for the discrepancy in AA simulations. Some of the differences in simulations do occur due to limitations in our current understanding of $O_3$'s impacts on antioxidants and grain quality which subsequently affects the assumptions made in the model construct and also the calibration. I have discussed this in sections 5.3 and 5.4 and direct the reader here in the new discussion section. However, there is an additional factor affecting the amino acid simulations which was not discussed prior, and this relates to the regressions used to convert grain protein to grain amino acids. The additional section in the discussion is as follows:

"While the grain methionine concentrations were reproduced well, the grain lysine concentrations were overestimated for the elevated $O_3$ treatment. It is also clear to see that the reduction in concentrations of both lysine and methionine was underestimated by the $DO_3SE$-CropN model. The AA concentrations were calculated using regressions linking protein concentrations to AAs from Liu et al. (2019), which were constructed using data from 48 field experiments from major wheat producing areas in China. Approximately 95% of wheat grown in China is winter wheat (United States Department of Agriculture, 2022), and most of the cultivars used to produce the regressions were winter wheat (Liu et al., 2019). However, the model was parameterised for Indian spring wheat. Given the differences between the growing conditions in India and China, and spring and winter wheat, deviations in simulations of lysine and methionine concentrations from the observed are to be expected. Additionally, Liu et al. (2019) did not include experiments with differing levels of $O_3$ in their treatments. For lysine, this has culminated in a much better simulation of the AA concentrations under ambient $O_3$ compared to the elevated treatment. For both lysine and methionine, using the regressions alone to convert grain protein to grain AA concentrations was not sufficient to account for the $O_3$ effect on grain quality. Additionally, there is currently a knowledge gap (discussed further in section 5.3) relating to our understanding of $O_3$'s effects on both antioxidants and grain quality, which affects not only the construction of the model but also its parameterisation. Suggestions for experiments which could reduce the knowledge gap for both modelling and understanding the effect of under $O_3$ exposure on grain protein and AAs are discussed in sections 5.3 and 5.4."

2. The study examines the critical issue of $O_3$ pollution and its impact on food security in India, highlighting the significance of this research given global nutrition challenges. The study could be improved by addressing potential regional variations in $O_3$ sensitivity within the context of the model's application to Indian wheat and exploring how this framework may be adapted for other significant wheat-producing regions experiencing comparable environmental stressors.

We agree with the reviewer that such a discussion would add great value to the manuscript. We believe the first comment: "addressing potential regional variations in $O_3$ sensitivity within the context of the model's application to Indian wheat" fits well with the current discussion section "Further work for understanding $O_3$ effects on wheat nutrition". Into this section we have added some remarks on the current wheat growing regions in India which are projected to experience the greatest $O_3$ effect on yield due to having the greatest modelled stomatal $O_3$ uptake, as these will likely overlap with the regions that will experience the greatest $O_3$ effect on nutrition. In this, we consider only models that have estimated stomatal $O_3$ uptake, as concentration-response studies have also been used to predict the spatial impact of $O_3$ on yields in India, but studies have shown that the areas with the greatest $O_3$ concentrations do not always overlap with those with the greatest yield losses due to the modifying effect of the environment on $O_3$ uptake (Pleijel, Danielsson and Broberg, 2022; Emberson et al., 2000). (e.g. If it is hot the stomata are likely closed and not taking up $O_3$).

The second part of the comment "exploring how this framework may be adapted for other significant wheat-producing regions experiencing comparable environmental stressors" is very interesting. In section 5.2 we had previously written: "The design of the antioxidant equations has several benefits which make it useful for further applications. Firstly, the structure of Eq. 1 means that it could be easily translated to other ROS mediated stressors, provided the corresponding equation parameters are identified, meaning the framework is flexible." Other ROS mediated stressors include high temperature and drought stress, which have been shown to also cause yield and protein reductions. Given that all the stressors cause similar effects on crop yield and quality, and are ROS mediated, it can be assumed that the mechanisms of reductions to yield and quality are similar and could be approximated using the same mechanism. In order to use such a mechanism, we would require a suitable proxy for measuring damage. In this study, we linked accumulated $O_3$ flux to antioxidant production. However, for drought stress it would not be as simple, due to the fact that N is taken up by the plant dissolved in water. In this case, the crop model would require suitable soil water algorithms to first simulate the effect on nutrient uptake, and to then simulate the effect of drought stress on increasing antioxidant production a suitable metric could be the duration and timing (e.g. pre- or post-anthesis) that the stressor occurs. Similarly, for heat stress it could be timing and duration of the stress. We incorporated these comments as follows:

"The design of the antioxidant equations has several benefits which make it useful for further applications. Firstly, the structure of Eq. 1 means that it could be translated to other stressors provided they have a similar mechanism of damage to $O_3$, meaning the framework is flexible. Drought and high temperature stress are good candidates for this framework as they are ROS mediated, like $O_3$, and cause a reduction in both grain yield and protein content (Broberg et al., 2015, 2023; Mariem et al., 2021). The effect of heat stress on antioxidant production, and hence grain quality, could be incorporated by modifying Eq. 1 and Fig. 2 to incorporate the duration (and potentially timing) of the stress as these are the key factors affecting grain yield under heat stress (Balla et al., 2019). For drought stress, the duration of the stress would be useful, but there would need to be an additional effect of drought on reducing nutrient uptake (as this affects grain quality) (Rijal et al., 2020; Faisal et al., 2017). The second benefit of the framework is that it is simple…"

3. The suggestion to combine nitrogen and protein assessments from leaves and stems, along with a deeper exploration of nitrogen allocation to antioxidants, is noteworthy. These efforts are expected to enhance model precision. It would be beneficial for the authors to delineate the

types of experimental data required to refine these aspects and to articulate specific hypotheses concerning the influence of antioxidant allocation on grain protein quality under $O_3$ stress.

We thank the reviewer for this suggestion and agree that this would then help future work to target the remaining uncertainties regarding $O_3$, antioxidant processes and $O_3$'s effects on crop nutrition. We believe that the reviewers' suggestions best fit in the final section "Further work for understanding $O_3$ effects on wheat nutrition" In this section we have two bullet points which originally vaguely summarised the kinds of information required to further develop our understanding: "3) To advance the antioxidant equations, and understand O3 effects on grain quality, an experiment measuring N and protein concentrations in the leaf and stem at anthesis, and harvest, should be conducted. The proportion of N associated with antioxidants under the same O3 treatments should also be obtained to improve mechanistic understanding of plant antioxidant response to O3 which can be used to further develop the model. 4) Relationships linking grain protein to grain AA concentrations should be investigated for how they change under the influence of O3. The modified equations could be integrated in the model so improve its ability to simulate AA concentrations under stress, and hence provide more trustworthy estimates of protein quality."

We improve on these two points by referencing the specific type of experiment that is required to obtain such data, which are $O_3$ exposure experiments. We don't distinguish whether these should be solardome, OTC or FACE as all would provide valuable information. We also specify greater detail on the kinds of data which should be obtained and which research questions the data will help to address. The improved text is as follows:

"3) To advance the antioxidant equations, and understand O3 effects on grain quality, an O3 exposure (e.g. FACE, OTC or solardome) experiment measuring total N and protein content, and N and protein concentrations in the leaf and stem at anthesis, and harvest stages under varying O3 treatments should be conducted. The proportion of N associated with specific antioxidants (such as glutathione and enzymatic antioxidants) under these O3 treatments should also be obtained to improve mechanistic understanding of plant antioxidant response to O3. This can be used to further develop the model, as it is anticipated that increased allocation of N to antioxidant production in leaves and stems under O3 stress reduces the N available for remobilisation to grains during grain filling, leading to a decrease in grain protein concentration and altered amino acid profiles.

4) From the same O3 exposure experiments, measurements of grain protein and AA concentrations for each O3 treatment should be collected to produce relationships linking the two and how the relationship changes under the influence of O3 to verify whether there is a trade-off between stress mitigation and nutritional quality. Such relationships could be integrated in the model to improve its ability to simulate AA concentrations under stress, and hence provide more trustworthy estimates of protein quality."

4. While the model accurately predicts yield loss, the discrepancies in amino acid concentration predictions indicate a need for further calibration. Additional validation steps, such as utilizing independent datasets or conducting field trials, may improve the credibility and generalisability of the model outputs.

We agree completely with the reviewer here. Unfortunately, there is limited availability of such data. To date, the only study that has investigated the effect of $O_3$ on amino acid concentrations

in wheat is that conducted by Dr Durgesh Singh Yadav, who generously provided his data and expertise for the development of the present model. We hope that this study will provide a modelling foundation, and useful suggestions for experimentalists and modellers alike so that in the future we may improve our understanding of $O_3$'s effects on crop nutrition further. We have incorporated remarks to this effect in the main manuscript as others may have similar questions.

We added the following at the beginning of the new section 5.2 (see response to comment 1): "To date, there is only one study (by Yadav et al. (2020)) that has investigated the effect of elevated $O_3$ on the AA concentrations of wheat . Data from this study was used to calibrate and evaluate the $DO_3SE$-CropN model, as well as test the framework for the AA simulations." Then at the end of the new section 5.2 we write: "Suggestions for more specific experiments which could reduce the knowledge gap for both modelling and understanding the effect of under $O_3$ exposure on grain protein and AAs are discussed in sections 5.3, 5.4 and 5.7. Nevertheless, it is clear that additional data on $O_3$'s effect on grain AA would be beneficial for not only model development, but improving confidence for modelling results."

The particular comment addressed in point 3 then link nicely to the additions here as well, as point 3 then provides ideas as to how these particular kinds of experiments would take place.

5. The authors emphasize the model's adaptability in simulating responses to various abiotic stressors. To enhance the manuscript, it would be beneficial to include examples of specific stressors, such as drought and heat, to which this framework could be adapted and discuss any preliminary adaptations made to expand its applicability.

I agree with the reviewer and believe I have now incorporated this in response to their comment 2, where I have included some remarks on how to incorporate drought and heat stress into the framework, with additional remarks for drought which will also affect nutrient uptake

**Response to RC3**

General comments:

This study extends the $DO_3SE$-CropN model to incorporate an O3-induced modification in nitrogen remobilization from the leaves and stems to the grains. The goal of the new mechanism is to improve the performance of the $DO_3SE$-CropN model in predicting the O3-driven reductions in wheat grain yield, protein, lysine and methionine concentrations in India. The study is novel, interesting and well-written. The model shows acceptable skill in predicting some variables (e.g. grain yield in single years as well as protein concentrations), whilst other outputs need further improvement. The authors report the limitations of the study clearly and concisely. I favour the publication of the manuscript following the modifications below:

Specific comments:

1) You use the concept of relative yield loss (RY loss) (e.g. Fig. 3b and Fig. S8) in a way that is not clear to me. The only definition that I was able to find was in the legend of Fig. S8: 'RY loss was calculated comparative to preindustrial O3 concentrations of 10 ppb (CLRTAP, 2017).' You should add some text to the main manuscript to define what is RY loss in the context of this study. Moreover, if RY loss is not measured but estimated, why is the x-axis of Fig 3b defined as 'observed'? Also, why is it important for $DO_3SE$-CropN to estimate correctly a trait like RY loss which was not observed here?

We thank the reviewer for bringing this to our attention. We have modified the manuscript to

incorporate an additional section at the end of the model development to explain the calculations for obtaining the relative yield loss. For completeness we also include an explanation of why we modelled the decrease in protein and amino acid concentrations rather than their relative values here. We then refer the reader to this section in the captions for Fig 3b and Fig S8. We feel that "observed" is still the correct term to use here as it is the RY loss calculated from the observational data, rather than simulations. The new section is as follows:

"2.5 Calculations of RY loss, and the decrease in protein and AA concentrations under $O_3$

For performing risk assessments of $O_3$ damage to crops, RY and RY loss (1-RY) are the commonly used response parameters, which quantify the magnitude of the crop yield loss under $O_3$ by comparing it to the corresponding pre-industrial value (~10 ppb) (see Eq. 5) (CLRTAP, 2017). Such risk assessments allow for the magnitude of the effects of $O_3$ on crop yields to be estimated (Emberson, 2020).

$$Relative\ yield\ (RY) = \frac{Yield\ under\ O_3\ treatment}{Yield\ under\ preindustrial\ O_3} \tag{5}$$

For the model simulations, the yield under preindustrial $O_3$ was extracted by performing a model run with a constant $O_3$ concentration of 10 ppb. While the yields under the $O_3$ treatment were obtained by running the model with the hourly experimental $O_3$ concentration data for the ambient and elevated (ambient + 20 ppb) treatments. To extract the yield under preindustrial $O_3$ concentrations for the experimental data, the yields for the ambient and elevated treatments were regressed against their M7 value. The regression was then used to calculate the expected yield at a preindustrial M7 of 10 ppb.

The calculations for obtaining the observed RY for the experimental data assume that the response of yield to increasing $O_3$ concentrations is approximately linear, which is verified in the literature (Pleijel, Danielsson and Broberg, 2022). However, the effect of $O_3$ on leaf and grain protein, and grain amino acids has received far less attention in the literature, and it is unknown if their response to increasing $O_3$ is also linear. These factors meant it was not possible to estimate preindustrial leaf and grain protein and grain AA concentrations. Instead, we focus on the reduction in leaf and grain protein, and grain AAs under the elevated, as compared to the ambient, $O_3$ treatment."

2) Fig 7: I suggest to add the cumulative O3 concentrations for the years 2017 and 2018 to the plot. If not possible, you can make a new plot with the cumulative O3 for both years (it could be added to the supplementary material if you prefer). It would be interesting to show to time-series comparison of accumulated O3 concentrations between the two years.

Thank you for this suggestion, we have produced a plot with the daily $O_3$ concentrations in parts per billion and overlayed the accumulated stomatal $O_3$ flux for each year to show the comparison between the concentrations and the amount of ozone the plant accumulates (differs depending on meteorology). We have added this to the supplementary as Figure S11, and in the text where we refer to Figure 7 we also refer the reader figure S11 to see a full view of the ozone data.

[Figure]

Technical corrections:

1) Fig 6-9: How did you calculate the differences in Temperature, O3, PPFD and net photosynthetic rate? Did you subtract the values of the year 2018 from the year 2017 or the opposite? In other words, which is your reference year? Please add this information to the figures' legends. The same is true for Figures S1-S7.

We thank the reviewer for spotting this, we had in the y axis as "2017 – 2018", but realise that this could be interpreted as "2017 to 2018" in terms of the spread of years, rather than the subtraction of the 2018 data from the 2017 data. We have modified all figure captions in the main text and supplementary to have an explicit statement of which year was subtracted from which.

We would like to extend our thanks to the reviewer for their suggestions which highlighted key areas of improvement for our manuscript. We feel the strength of the paper has been substantially improved by their suggestions.

**Response to CC1**

General comments: The authors have submitted an extremely well-written manuscript to the TOAR-II Community Special Issue, and the paper has already received two thorough reviews from the anonymous referees. I find the paper and its conclusions to be consistent with the findings of previous TOAR work and I only have a few minor comments, listed below.

Line 70-72 The first phase of TOAR was only able to find publicly available ozone data at 3 sites in India (Mills et al., 2018), and only two of these were in the wheat growing areas of northern India. Given that observations are so sparse, how confident are you that India has some of the highest ozone levels with regards to crops? Have additional ozone observations become available at more sites across India since the first phase of TOAR? Is your assessment based on models?

We thank the editor for commenting on this, this paragraph has now had substantial edits based on these remarks and those of reviewer 1 which we hope to make it clearer which information

comes from observations, which come from model projections and the limitations of the associated TOAR1 data. We highlight these changes in the screenshot below:

**1.2 O₃ pollution in India**

Ground level O₃ is a secondary pollutant, formed when precursor gases (predominantly volatile organic compounds and nitro oxides) react in the presence of ultraviolet light (Fowler et al., 2008). T the first phase of TOAR, identified that South Asia, and in particular India, experience some of the highest O₃ burdens of any region or country worldwide, though this analysis was limited by the availability of O₃ concentration data for India (Mills et al., 2018b; Emberson, 2020). These high O₃ burdens occur due to increasing pre-cursor emissions and insufficient pollution control measures (Archibald et al., 2020; Singh, Dey and Knibbs, 2023; Wang et al., 2023; Elshorbany et al., 2024). Atmospheric chemistry and climate models have found that geographically, the highest O₃ concentrations in India occur in the northern part of the country and the Indo-Gangetic planes (IGP), where the majority of wheat is grown (Rathore, Gopikrishnan and Kuttippurath, 2023; Lu et al., 2018; Ministry of Agriculture & Farmers Welfare, 2022). In the future, the changing climate will affect O₃ concentrations, with model projections agreeing that climatic conditions across the north of India  will favour greater O₃ production (Stevenson et al., 2013; Li et al., 2022a; Kumar et al., 2018). Using a Nested Regional Climate Model with Chemistry, (Kumar et al. ( 2018) projected that O₃ concentrations across India will rise under RCP 8.5, while remaining comparable to current levels under RCP 6.0. For the dry, wheat growing season, the authors projected that O₃ concentrations across the IGP will increase under both RCP 6.0 and RCP 8.5, with a much larger increase under RCP 8.5." This is a critical finding given the majority of wheat is grown in the north of India, across the IGP (Ministry of Agriculture & Farmers Welfare, 2022).

Line 69 Nitrous oxide should be nitrogen oxides

We thank the editor for spotting this and have corrected the manuscript accordingly

Line 244 29016 should be 2016

We thank the editor for spotting this mistake and have corrected it

Line 410 Meteorology's doesn't seem to be the right work. Would "meteorological conditions" be more suitable?

Yes this is much more suitable, we have edited the manuscript to reflect this

We thank the editor for their suggestions, which have made the manuscript more robust, particularly with regards to section 1.2 where we discuss the current state of the literature. We feel the strength and accuracy of this section has been improved as a result of their feedback.